# Interpretable Hypergraph Neural Additive Networks

## Abstract

Hypergraph neural networks have emerged as a powerful framework for learning from higher-order structured data, where relationships among entities extend beyond pairwise connections. However, most current hypergraph neural networks are black-boxes that rely on post-hoc explanation methods to provide model insights. Such post-hoc explanations can be unreliable in high-stakes scenarios and knowledge discovery tasks. We introduce an inherently interpretable hypergraph neural additive network (HGNAN), an extension of generalized additive models that facilitates interpretability in complex, higher-order relational learning settings. HGNAN provides clear visualizations of both global and local behaviors at the node and hyperedge levels while preserving the expressive power of hypergraphs. We evaluate HGNAN on node classification and hyperedge prediction across various datasets, achieving competitive performance compared to state-of-the-art methods. HGNAN also significantly outperforms existing approaches in recovering missing reactions in metabolic networks, while offering interpretable biological insights into metabolic processes.

## 1 Introduction

Hypergraphs are effective in capturing complex interactions among multiple entities and have been widely applied across various domains, such as metabolic, ecological, and social networks (Battiston et al., 2020; Chen et al., 2023; Grilli et al., 2017; Zhu et al., 2018). Unlike traditional graphs where each edge connects only two nodes, a hyperedge can link any number of nodes, offering greater expressive power and flexibility in modeling multidimensional real-world systems (Berge, 1984). Recent advances have led to significant success in tasks like node classification and hyperedge prediction (Gao et al., 2020; Schölkopf et al., 2007). In particular, hypergraph neural networks (HGNNs) have emerged as a state-of-the-art method for representation learning, effectively leveraging higher-order structural patterns (Bai et al., 2021; Feng et al., 2019). However, like many deep learning models, HGNNs operate as black boxes, providing limited insight of their decision-making processes. This lack of transparency raises concerns about trust, restricts their adoption in high-stakes applications, and impedes scientific knowledge discovery.

Post-hoc explainability techniques, such as HyperEX (Maleki et al., 2023) and SHypX (Su et al., 2024), have been proposed to open the black box. Inspired by graph-based counterparts such as GNNExplainer (Ying et al., 2019), PGExplainer (Luo et al., 2020), and XGNN (Yuan et al., 2020), both methods identify sub-hypergraphs that maximize mutual information with the model's output. These extracted patterns are designed to enhance the predicted probability for a specific class, offering model-level explanations. However, both approaches rely on an auxiliary explanatory model to interpret a trained HGNN. Such explanations can be unreliable and may even undermine trust in the model's decisions (Rudin, 2019). Recently, inherently interpretable graph neural networks have been introduced by using prototype reasoning (Dai & Wang, 2025) and generalized additive models (Bechler-Speicher et al., 2024). Nevertheless, to the best of our knowledge, interpretable HGNNs remain unexplored.

In this paper, we introduce an inherently interpretable neural network designed for hypergraph learning tasks, named hypergraph additive neural network (HGNAN) (see Figure 1). HGNAN

integrates the principles of neural additive models (NAMs) (Agarwal et al., 2021) with HGNN architectures. Specifically, it adds distance-based weights into NAMs to learn structural information, adopting the message-passing mechanism of HGNNs to model higher-order relationships. HGNAN can provide clear interpretations and visualizations of both global and local behaviors at the node and hyperedge levels. Concretely, for both tasks HGNAN provides feature-level interpretations. In addition, it offers neighbor-level explanations for node-level tasks and distance-based interpretations for hyperedge-level tasks. Our experiments show that HGNAN achieves performance comparable to various baselines in both node and hyperedge prediction tasks while providing interpretations of exact decision-making process of the underlying model. HGNAN also establishes a new state-of-the-art in identifying missing reactions in genome-scale metabolic networks, and its interpretability enables the discovery of novel biological insights that enhance our knowledge of metabolic processes.

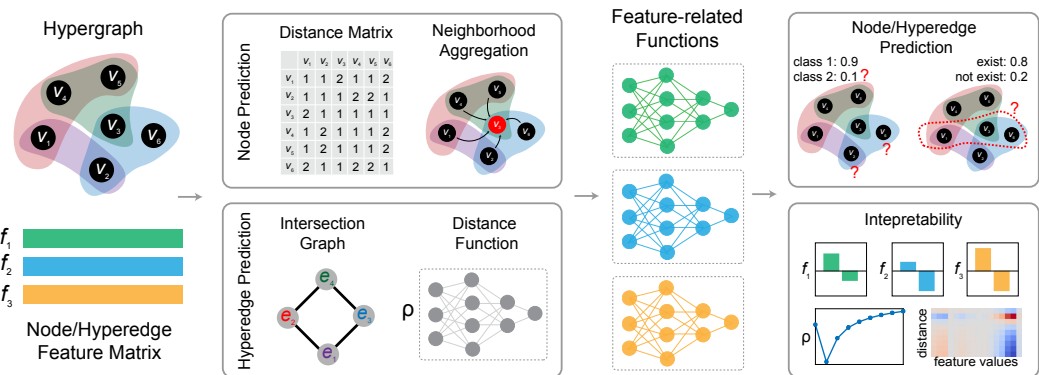

Figure 1: Overview of HGNAN. HGNAN takes a hypergraph and a feature matrix as input (leftmost column) and processes them through two independent components: one that captures distance-related information (mid-left column) and another that captures feature-related patterns (mid-right column). For node prediction, HGNAN computes a distance matrix and performs neighborhood aggregation. For hyperedge prediction, it builds an intersection graph and applies a distance function between nodes. As in NAMs, each feature-related function operates only on a single feature to ensure interpretability.

## 2 RELATED WORKS

**Hypergraph neural networks.** Hypergraph neural networks (HGNNs) were first introduced by Feng et al. (2019) via a spectral formulation based on the hypergraph Laplacian. Since then, many architectures have been proposed, including hyperedge-reduction methods such as HAN (Chen et al., 2020) and HyperGCN (Agarwal et al., 2021), set-function-based models such as AllSetTransformer (Chien et al., 2021), and more expressive variants such as UniGNN (Huang & Yang, 2021), HGNN+ (Gao et al., 2023), and ED-HNN (Wang et al., 2023a). These models differ in how they encode hyperedges and propagate messages, but they are all designed as black-box predictors and offer little inherent interpretability.

**Post-hoc explainability for (hyper)GNNs.** To explain black-box GNNs, post-hoc methods such as GNNExplainer (Ying et al., 2019), PGExplainer (Luo et al., 2020), and XGNN (Yuan et al., 2020) learn features or subgraph structures that are important for a given prediction via masking on pretrain GNNs. Recently, two post-hoc explainers tailored to HGNNs have been proposed, which adapt these ideas to hypergraphs. HyperEX (Maleki et al., 2023) is a model-agnostic framework that computes node-hyperedge pair importance scores and extracts sub-hypergraphs as explanations for node predictions. SHypX (Su et al., 2024) further extends this idea by sampling faithful explanation sub-hypergraphs at instance level and clustering them into global concept-level explanations for a trained HGNN. While effective in many settings, such post-hoc approaches do not constrain the underlying model itself and their explanations are not guaranteed to faithfully reflect its true reasoning process, which can undermine trust in high-stakes applications (Rudin, 2019).

**Interpretable-by-design additive and graph models.** In contrast, additive models aim to be interpretable by design. Classical generalized additive models (GAMs) (Hastie & Tibshirani, 1986) represent predictions as a collection of univariate shape functions over individual features, which can be inspected via simple 2D plots. Neural additive models (NAMs) (Agarwal et al., 2021) extend this idea by learning a small neural networks for each feature, combining non-linear expressiveness of neural networks with decomposable architecture of GAMs. Graph neural additive networks (GNANs) (Bechler-Speicher et al., 2024) further extends this idea to graphs by combining feature-wise subnetworks with distance-aware structural aggregation. Beyond additive models, several GNN variants also introduce interpretability directly into the architecture, such as GSAT (Miao et al., 2022), which uses an information-bottleneck-motivated stochastic attention to select sparse explanatory subgraphs, and concept-whitening GNNs (Proietti et al., 2023), which align hidden channels with human-understandable concepts. These works demonstrate that GNNs can be made inherently interpretable without a separate explainer. However, interpretable-by-design additive modeling on hypergraphs with HGNNs has yet to be explored. HGNAN follows this interpretable-by-design philosophy and, to the best of our knowledge, is the first hypergraph neural network that adopts a NAM-style additive structure to provide decomposable explanations at the feature, node-neighborhood, and hyperedge-distance levels.

## 3 METHODS

A hypergraph generalizes the concept of traditional graphs by allowing each hyperedge to connect an arbitrary subset of nodes, rather than being limited to pairwise connections. Formally, a hypergraph $\mathcal{H} = \{\mathcal{V}, \mathcal{E}\}$ with $n$ nodes and $m$ hyperedges consists of a set of nodes $\mathcal{V} = \{v_1, v_2, \ldots, v_n\}$ and a set of hyperedges $\mathcal{E} = \{e_1, e_2, \ldots, e_m\}$, where each hyperedge $e_j \in \mathcal{E}$ is a non-empty subset of $\mathcal{V}$, i.e., $e_j \subseteq \mathcal{V}$. The structural relationships in a hypergraph can be represented using an incidence matrix $\mathbf{H} \in \mathbb{R}^{n \times m}$, where each entry $\mathbf{H}_{ij} = 1$ if node $v_i$ is contained in hyperedge $e_j$, and it is equal to zero otherwise. Given a hypergraph $\mathcal{H}$, its intersection graph is defined upon hyperedge adjacency. Each node in the intersection graph represents a hyperedge in $\mathcal{E}$, and an edge is placed between two nodes if their corresponding hyperedges share at least one common node in $\mathcal{V}$. In other words, an edge between two nodes $e_i$ and $e_j$ exists if $e_i \cap e_j \neq \emptyset$.

Building on the formulation of hypergraphs, we present a novel interpretable hypergraph neural additive network (HGNAN), which consists of two parts: HGNAN-node for node-level prediction and HGNAN-edge for hyperedge-level prediction. Both take the incidence matrix $\mathbf{H}$ and node features as input to learn feature shape functions that capture the effects of individual features. The key distinction is that HGNAN-node leverages the hypergraph structure to perform neighborhood-level aggregation, whereas HGNAN-edge learns a separate distance function defined over higher-order intersection graphs.

### 3.1 DISTANCE ON HYPERGRAPHS

Given a hypergraph $\mathcal{H} = \{\mathcal{V}, \mathcal{E}\}$, a pair of hyperedges $e_i, e_j \in \mathcal{E}$ is called $s$-adjacent if they share at least $s$ nodes in common, i.e., $|e_i \cap e_j| \geq s$. This allows us to extend intersection graphs to $s$-intersection graphs by defining the $s$-hyperedge adjacency matrix $\mathbf{A}(s)$ as

$$\mathbf{A}(s)_{ij} = \mathbf{1}[|e_i \cap e_j| \geq s]$$

where $\mathbf{A}(s) \in \{0, 1\}^{m \times m}$. For any pair of hyperedges $e_i, e_j \in \mathcal{E}$, we define the $s$-distance between the two hyperedges as $d_s(e_i, e_j) = \{\min \alpha \in \mathbb{Z}^+ \mid \mathbf{A}^\alpha(s)_{ij} > 0\}$, where $\alpha$ can be interpreted as the shortest distance between $e_i$ and $e_j$. If no such $\alpha$ exists, we set $d_s(e_i, e_j) = \infty$. In other words, $d_s(e_i, e_j)$ is the minimum number of hops required to connect $e_i$ and $e_j$ via a chain of $s$-adjacent hyperedges. Additionally, the $s$-distance between two nodes $v_i, v_j \in \mathcal{V}$ can be defined as

$$d_s(v_i, v_j) = 1 + \min_{v_i \in e_p, \, v_j \in e_q} d_s(e_p, e_q). \tag{1}$$

If no connected pair of hyperedges that contain $v_i$ and $v_j$ can be found, we set $d_s(v_i, v_j) = \infty$. If $d_s(v_i, v_j) \leq 2$, we consider that node $v_i$ is in the neighborhood of node $v_j$, i.e. $v_j \in \mathcal{N}_i$.

This implies that nodes on the same hyperedge and those one hop away are neighbors. We denote the neighborhood under $s$-adjacency by $\mathcal{N}_i^s = \{v_j \in \mathcal{V} : d_s(v_i, v_j) \leq 2\}$. We denote $s_{\max}$ as the highest order of adjacency considered and is introduced as a tunable hyperparameter, enabling the model to learn higher-order adjacency from the hypergraph. In practice, we consider $s \in \{1, \ldots, s_{\max}\}$, where $s_{\max} \in \mathbb{Z}^+$ controls the maximum order of adjacency.

## 3.2 Node Prediction Tasks

HGNAN-node generalizes NAMs to hypergraph-structured data by learning a set of feature-wise shape functions $\{f_k\}_{k=1}^p$, where $p$ denotes the number of input features. Each shape function $f_k$ transforms the $k^{\text{th}}$ feature independently across a node's neighborhood and contributes to a refined representation via neighborhood-level aggregation. This design effectively suppresses noise from irrelevant neighbors, mitigates oversmoothing from repeated mixing, and enhances both memory and computational efficiency. Let $\mathbf{x}_i \in \mathbb{R}^p$ denote the original features and $\mathbf{h}_i^s \in \mathbb{R}^p$ the updated embedding for node $i$ under $s$-adjacency. The $k^{\text{th}}$ entry of the embedding $\mathbf{h}_i^s$ using HGNAN-node is computed as

$$[\mathbf{h}_i^s]_k = \sum_{v_j \in \mathcal{N}_i^s} \frac{a_{ij}}{\#d_s(v_i, v_j)} f_k([\mathbf{x}_j]_k) + \lambda \|\mathbf{a}\|_1, \tag{2}$$

where $\mathcal{N}_i^s$ denotes the neighborhood of node $i$ under $s$-adjacency (the definition for neighboorhood of node $v_i$ is the same as described in Section 3.1), and $f_k(\cdot)$ is a feature-dependent shape function associated with the $k^{\text{th}}$ feature. Each $f_k$ is implemented as an MLP mapping $\mathbb{R} \rightarrow \mathbb{R}$. A neighbor weight $a_{ij}$ is assigned to each neighbor $j$ of the target node $i$ and is calculated using a small neural network similar to graph attention network (Velickovic et al., 2018). The vector $\mathbf{a}_i = (a_{ij})_{j \in \mathcal{N}_i^s}$ denotes all neighbor weights for node $i$. To encourage neighborhood-level sparsity, we apply an L1-norm to neighbor weights. The term $\#d_s(v_i, v_j)$ measures the number of nodes sharing the same distance away from node $i$ under $s$-adjacency, serving as a normalization factor. HGNAN-node follows the formulation of NAMs and GNANs, where $[\mathbf{h}_i^s]_k$ is a linear combination of the transformed features $f_k([\mathbf{x}_j]_k)$, representing the aggregated contribution of the $k^{\text{th}}$ channel at node $i$.

For node-level prediction, we introduce an $s$-invariant weight vector $\mathbf{w}$ which learns how each entry contributes to the prediction under $s$-adjacency $[\mathbf{h}_i^s]$, i.e.,

$$[\mathbf{h}_i^s] = \sum_{k=1}^p w_k [\mathbf{h}_i^s]_k = \sum_{v_j \in \mathcal{N}_i^s} \frac{a_{ij}}{d_s(v_i, v_j)} \sum_{k=1}^p w_k f_k([\mathbf{x}_j]_k), \tag{3}$$

where $w_k$ is the $k^{\text{th}}$ element of $\mathbf{w}$, which acts as a learnable weight for feature $k$ and is normalized using softmax so that $\sum_{k=1}^K w_k = 1$. We have $\mathbf{w} \in \mathbb{R}^p$. This $w_k$ could act as a relative feature importance for feature $k$ for the whole task. The final embedding for node $i$ is a sum weighted by learnable parameter $\beta$ over $\{[\mathbf{h}_i^s]\}_{s=1}^{s_{\max}}$, which are embeddings learned under all high-order adjacency:

$$[\mathbf{h}_i] = \sum_{s=1}^{s_{\max}} \beta_s [\mathbf{h}_i^s], \tag{4}$$

where $\boldsymbol{\beta} \in \mathbb{R}^{s_{\max}}$ and each $\beta_s$ is learned. Eq (4) allows the model to flexibly learn each neighbor's contribution to node $i$ by considering both high-order adjacency and feature-level importance. This representation can then be passed through a sigmoid or softmax activation function to flexibly adjust for either binary or multiclass classification or regression. Furthermore, HGNAN-node can be naturally extended to hypergraph-level prediction by pooling the node-level aggregated scores into a single hypergraph-level representation $\mathbf{h}$, i.e.,

$$\mathbf{h} = \sum_{i=1}^N [\mathbf{h_i}] \tag{5}$$

where $\mathbf{h} \in \mathbb{R}$ for binary tasks or $\mathbf{h} \in \mathbb{R}^C$ for $C$-class classification.

### 3.3 Hyperedge Prediction Tasks

Hyperedge prediction methods usually first compute node embeddings for individual nodes and then combine those node embeddings through a pooling layer to get an embedding for the hyperedge. However, this pooling step can lead to several issues. For example, max-pooling may cause non-smoothness and gradient instability, while average pooling can lead to gradient dilution and reduced receptive-field gradients (Boureau et al., 2010). Additionally, it may compromise model interpretability by offering node-level explanations for hyperedge predictions, rather than providing explanations directly from the hyperedge-level perspective. More importantly, for most node classification datasets, they are highly homophilic. 0-hop and 1-hop neighbors in these datasets usually covers most of the nodes. In this away, deploying neighborhood aggregation can cover most of the valuable information. In contrast, for hyperedge prediction tasks, they are highly heterophilic, which means that neighborhood aggregation cannot capture enough information. To address these issues, we propose HGNAN-edge, which leverages the concept of $s$-intersection graphs and uses overall aggregation for hyperedge prediction. See Appendix B for more details.

In HGNAN-edge, we first transform the hypergraph into its corresponding $s$-intersection graph, and then generate hyperedge embeddings directly from features of the nodes associated with each hyperedge. By performing pooling on raw node features, this new method allows us to design interpretable hyperedge embedding based on prior knowledge. As a result, the model can provide hyperedge-level interpretation, which preserves better interpretability compared to those that pool after node embedding. This new formulation turns hyperedge prediction into a node prediction task on the $s$-intersection graph, thereby HGNAN-edge can easily adapt GNAN to get hyperedge embeddings and make predictions.

Let $\mathbf{x}_l \in \mathbb{R}^p$ denote the original feature vector of node $v_l$. The initial embedding for hyperedge $e_i$ is $\mathbf{y}_i = \text{Pool}(\{\mathbf{x}_l : v_l \in e_i\})$, where $\text{Pool}(\cdot)$ denotes a pooling function, such as average pooling. Thus $\mathbf{y}_i \in \mathbb{R}^p$. HGNAN-edge computes the $k^{\text{th}}$ entry of the refined embedding $\mathbf{g}_i^s \in \mathbb{R}^p$ on the $s$-intersection graph as follows:

$$[\mathbf{g}_i^s]_k = \sum_{j=1}^m \frac{1}{d_s(e_i, e_j)} \, \rho_s \left( \frac{1}{1 + d_s(e_i, e_j)} \right) f_k \left( [\mathbf{y}_j]_k \right), \tag{6}$$

where $f_k(\cdot)$ as defined in Eq (2) is a feature-dependent shape function corresponding to the $k^{\text{th}}$ feature, and $\rho_s(\cdot)$ is a distance-based weighting function that captures the cumulative influence of hyperedges at varying distances from $e_i$ on the $s$-intersection graph. Each $f_k : \mathbb{R} \to \mathbb{R}$ and $\rho_s : \mathbb{R} \to \mathbb{R}$ is parameterized by an MLP. To avoid division by zero, we add 1 to each distance value in the denominator. Finally, we compute the refined hyperedge representation using the same aggregation strategy as HGNAN-node, i.e.,

$$[\mathbf{g}_i^s] = \sum_{k=1}^p w_k [\mathbf{g}_i]_k = \sum_{j=1}^m \frac{1}{d_s(e_i, e_j)} \, \rho_s \left( \frac{1}{1 + d_s(e_i, e_j)} \right) \sum_{k=1}^p w_k f_k \left( [\mathbf{y}_j]_k \right), \; [\mathbf{g}_i] = \sum_{s=1}^{s_{\max}} \beta_s [\mathbf{g}_i^s], \tag{7}$$

where $w_k$ and $\beta_s$ are learnable weights. We have $\mathbf{w} \in \mathbb{R}^p$ and $\boldsymbol{\beta} \in \mathbb{R}^{s_{\max}}$. Similar to HGNAN-node, Eq (7) can also be interpreted through two independent parts: a distance-related part $\sum_{j=1}^m \frac{1}{d_s(e_i, e_j)} \rho_s(\frac{1}{1+d_s(e_i, e_j)})$ and a feature-related part $\sum_{k=1}^p w_k f_k([\mathbf{y}_j]_k)$. However, unlike HGNAN-node, which employs neighborhood-level aggregation in the distance-related part, HGNAN-edge performs graph-level aggregation using a distance-based weight function $\rho_s(\cdot)$.

Note that for most existing hyperedge prediction datasets, HGNAN-edge requires negative sampling, meaning that we need to generate hypothesized hyperedges (also called negative hyperedges) from existing hypergraph. However, when we calculate the distances, we only use the information of the existing hypergraph structure without taking the new generated negative hyperedge into consideration. Specifically, for a generated negative sample $\tilde{e}_i$, we first identify the positive hyperedges it connects to and then calculate the distance based on those connections. Let $E_i$ and $E_j$ denote the sets of positive hyperedges connected to the negative hyperedges $\tilde{e}_i$ and $\tilde{e}_j$, respectively. The distance between the two sets $E_i$ and $E_j$ is defined as $d_s(E_i, E_j) = \min_{e_i, e_j} d_s(e_i, e_j)$ such that $e_i \in E_i, e_j \in E_j$. Accordingly, the

distance between two negative hyperedges $\tilde{e}_i$ and $\tilde{e}_j$ is defined as $d_s(\tilde{e}_i, \tilde{e}_j) = d_s(E_i, E_j) + 2$. If one hyperedge is positive and the other is negative, the distance generalizes naturally as $d_s(\tilde{e}_i, e_j) = d_s(E_i, e_j) + 1$.

### 3.4 TRAINING PIPELINE

HGNAN adopts a unified supervised learning pipeline for both node- and hyperedge-level tasks. Given node features $X \in \mathbb{R}^{n \times p}$ and incidence matrix $H$, we first construct the node $s$-adjacency and the hyperedge $s$-intersection graph. For node prediction, HGNAN-node directly uses $(X, H)$ and produces node embeddings $\{h_i\}_{i=1}^n$ through feature-wise shape functions and neighborhood aggregation; these embeddings are fed into a linear prediction head and optimized using cross-entropy or binary cross-entropy, following standard settings used in HGNN and AllSet. For hyperedge prediction, we compute hyperedge features $Y = \{\text{Pool}(\{x_l : v_l \in e_i\})\}_{i=1}^m$ (Section 3.3), and HGNAN-edge applies the same additive mechanism over the $s$-intersection graph to generate hyperedge embeddings $\{g_e\}_{e=1}^m$, which are passed to a prediction head and trained with binary cross-entropy for hyperedge existence prediction. Distances are always computed from the original hypergraph structure and are independent of negative sampling. All experiments use the same train/validation/test splits and optimization settings as the baselines, and the entire pipeline is end-to-end differentiable.

## 4 EXPERIMENTS

Our experiment aims to address the following two questions: (1) Can HGNAN achieve performance comparable with black-box counterparts in terms of node prediction (Section 4.1) and hyperedge prediction (Section 4.2)? (2) What do the interpretations from HGNAN-node and HGNAN-edge look like (Section 4.3)?

### 4.1 NODE CLASSIFICATION

We compare HGNAN-node with six hypergraph learning methods on four node classification tasks. Specifically, we compare to the following baselines: HGNN (Feng et al., 2019), HyperGCN (Yadati et al., 2019), AllDeepSets (Chien et al., 2021), AllSetTransformer (Chien et al., 2021), UniGCNII (Huang & Yang, 2021), ED-HNN (Wang et al., 2023a) and PhenomNN (Wang et al., 2023b). These baselines represent diverse architectural designs for modeling high-order relationships and have demonstrated strong performance on node classification tasks. We also compared with MLP, which does not utilize any hypergraph structure to validate that the model learns and benefits from structural information. We use three widely used homophilic hypergraph-level node classification datasets: Cora (Sen et al., 2008), Citeseer(Giles et al., 1998), Zoo (Forsyth, 1990), Mushroom (mus, 1981), and NTU2012 (Chen et al., 2003). We also tested on Pokec and Actor, two heterophilic datasets introduced by (Li et al., 2025). Details about these datasets are available in Appendix A. Each dataset is randomly split into training, validation, and test sets with a 2:1:1 ratio, and this process is repeated 10 times using different random seeds. To ensure a fair comparison, all models are trained and evaluated using the same data splits and random seeds. We report the mean and standard deviation of the test accuracy across these 10 runs.

Table 1 shows the test accuracy of HGNAN-node and baselines across six different datasets. HGNAN-node achieves accuracy comparable to that of baseline methods. Notably, it outperforms all baselines on two heterophilic node classifications datasets. While some baselines slightly outperform HGNAN-node on homophilic datasets, the performance gap remains relatively small. Crucially, unlike these black-box models, HGNAN offers the added advantage of providing glass-box view of the decision making process, making it a compelling choice in scenarios where both accuracy and transparency are essential.

### 4.2 HYPEREDGE PREDICTION

Genome-scale metabolic models (GEMs) are essential tools for predicting cellular metabolism and physiological states in organisms (Fang et al., 2020). However, due to incomplete

Table 1: Comparison of test accuracy (mean ± standard deviation) between HGNAN-node and baselines across node classification datasets. Bold indicates the highest test accuracy.

| Method | Zoo | Mushroom | NTU2012 | Cora | Citeseer | Pokec | Actor | Avg. Rank |
|---|---|---|---|---|---|---|---|---|
| MLP | $0.887 \pm 0.052$ | $0.965 \pm 0.006$ | $0.853 \pm 0.012$ | $0.753 \pm 0.014$ | $0.714 \pm 0.010$ | $0.580 \pm 0.019$ | $0.827 \pm 0.004$ | 7.0 |
| AllDeepSets | $0.942 \pm 0.042$ | $\mathbf{0.999 \pm 0.001}$ | $0.876 \pm 0.014$ | $0.769 \pm 0.015$ | $0.696 \pm 0.013$ | $0.567 \pm 0.008$ | $0.838 \pm 0.003$ | 5.7 |
| AllSetTransformer | $\mathbf{0.973 \pm 0.032}$ | $\mathbf{0.999 \pm 0.001}$ | $0.890 \pm 0.011$ | $0.784 \pm 0.016$ | $0.723 \pm 0.013$ | $0.572 \pm 0.010$ | $0.836 \pm 0.002$ | 3.0 |
| ED-HNN | $0.950 \pm 0.035$ | $0.998 \pm 0.002$ | $0.895 \pm 0.013$ | $\mathbf{0.801 \pm 0.018}$ | $\mathbf{0.729 \pm 0.016}$ | $0.618 \pm 0.020$ | $0.856 \pm 0.006$ | **2.6** |
| HGNN | $0.957 \pm 0.022$ | $0.998 \pm 0.001$ | $0.872 \pm 0.014$ | $0.787 \pm 0.012$ | $0.714 \pm 0.010$ | $0.553 \pm 0.014$ | $0.744 \pm 0.004$ | 5.7 |
| HyperGCN | $0.423 \pm 0.000$ | $0.482 \pm 0.000$ | $0.796 \pm 0.033$ | $0.775 \pm 0.020$ | $0.674 \pm 0.009$ | $0.538 \pm 0.014$ | $0.630 \pm 0.000$ | 8.7 |
| UniGCNII | $0.950 \pm 0.048$ | $\mathbf{0.999 \pm 0.001}$ | $0.893 \pm 0.016$ | $0.782 \pm 0.016$ | $0.715 \pm 0.011$ | $0.570 \pm 0.018$ | $0.828 \pm 0.003$ | 4.3 |
| Phenom | $0.968 \pm 0.000$ | $0.993 \pm 0.001$ | $\mathbf{0.898 \pm 0.008}$ | $0.799 \pm 0.005$ | $0.720 \pm 0.003$ | $0.594 \pm 0.002$ | $0.827 \pm 0.003$ | 3.4 |
| HGNAN-node (ours) | $0.953 \pm 0.030$ | $\mathbf{0.999 \pm 0.001}$ | $0.890 \pm 0.011$ | $0.779 \pm 0.020$ | $0.718 \pm 0.011$ | $\mathbf{0.634 \pm 0.012}$ | $\mathbf{0.857 \pm 0.004}$ | 3.0 |

knowledge of metabolic processes, even well-curated GEMs often contain knowledge gaps, such as missing reactions. Predicting these missing reactions can be naturally framed as a hyperedge prediction task, where nodes represent metabolites and hyperedges correspond to reactions (Chen et al., 2023; Chen & Liu, 2024). We compare HGNAN with the state-of-the-art hyperedge prediction methods including HyperSAGNN (Zhang et al., 2020), NHP (Yadati et al., 2020) and CHESHIRE (Chen et al., 2023), which have demonstrated strong empirical performance in recovering missing reactions in metabolic networks. We evaluate on four GEMs from the BiGG database (King et al., 2016): iAF1260b, iJR904, iJR904 and iYO844. Appendix B gives a detailed introduction to these datasets.

Since BiGG models do not have inherent features for metabolites, we design a feature generation process by leveraging molecular fingerprints MACCS Keys(RDK, 2025) to represent the metabolites (see Appendix B). To perform hyperedge prediction, negative sampling is required. For a given reaction in a GEM, a negative reaction is generated by replacing half of its metabolites with random metabolites drawn from a metabolite pool. Other experiment setups are identical to node prediction. We report the mean and standard deviation of the test accuracy across these 10 runs. Additional metrics and further details on the tuning process and parameter search space are available in Appendix C.

Table 2 summarizes the performance of various hyperedge prediction models on four GEM datasets. HGNAN consistently outperforms all state-of-the-art methods across all datasets, achieving the highest mean accuracy with notable margins. Notably, it outperforms the second-best method by over 10% on iAF1260b, iJR904, and iSB619. These results underscore the effectiveness of HGNAN in identifying missing reactions within complex metabolic networks. In addition to its strong predictive performance, HGNAN offers interpretability by linking metabolite chemical structures to prediction outcomes, providing valuable insights into the underlying biochemical mechanisms. An additional experiment in Appendix D on a synthetic hyperedge prediction dataset further validates our model's ability to recover the structural information.

Table 2: Comparison of hyperedge prediction accuracy (mean ± standard deviation) across four BiGG GEM datasets: iAF1260b, iJR904, iSB619, and iYO844. Bold highlights the best result for each dataset. HGNAN-edge outperforms all baseline models across all datasets.

| Method | iAF1260b | iJR904 | iSB619 | iYO844 |
|---|---|---|---|---|
| CHESHIRE | $0.834 \pm 0.050$ | $0.732 \pm 0.068$ | $0.730 \pm 0.038$ | $0.893 \pm 0.047$ |
| NHP | $0.732 \pm 0.076$ | $0.690 \pm 0.090$ | $0.687 \pm 0.055$ | $0.747 \pm 0.043$ |
| HyperSAGNN | $0.730 \pm 0.075$ | $0.753 \pm 0.056$ | $0.729 \pm 0.162$ | $0.708 \pm 0.045$ |
| HGNAN-edge (ours) | $\mathbf{0.935 \pm 0.069}$ | $\mathbf{0.958 \pm 0.026}$ | $\mathbf{0.977 \pm 0.008}$ | $\mathbf{0.952 \pm 0.181}$ |

### 4.3 INTERPRETABILITY

HGNAN provides a glass-box view of its decision-making process. According to Eqs (4) and (7), our model can be fully explained by two parts: the feature-related part and the distance-related part. The feature-related part, or Feature Contribution Score in

the context, is $w_k f_k([x_j]_k)$, which reflects how the input features influence the output embedding. The distance-related part differs between HGNAN-node and HGNAN-edge. For HGNAN-node, it is the neighborhood weight $a_{ij}$, while for HGNAN-edge, there are distance functions $\{\rho_s(\cdot)\}_{s=1}^{s_{max}}$, which adjust the influence of features with learnt high-order structural information. We can also learn relative feature importance for the whole task from $w_k$ (see Appendix E).

**Node-level interpretation.** HGNAN-node can offer node-level interpretation for node prediction tasks. Since we deploy neighborhood-level aggregation in HGNAN-node, the feature contribution scores $\{w_k f_k\}_{k=1}^p$, provides a global explanation of how each input feature contributes to the predictions. We use the Zoo dataset as an example. It contains 101 animals described by 15 binary features (e.g., whether an animal has feathers) and 1 continuous feature. Animals are grouped into seven classes: "Mammal", "Bird", "Reptile", "Fish", "Amphibian", "Bug", and "Invertebrate". Hyperedges connect animals that share common attributes. The goal is to predict the class to which each animal belongs.

To interpret the contribution of each binary feature to the prediction of the "Mammal" class, we plot the learned values $w_k f_k(1)$ and $w_k f_k(0)$ for each feature $k$, as shown in Figure 2. The sign of $w_k f_k(x_k), x_k \in \{0, 1\}$ indicates whether the presence or absence of a feature increases or decreases the likelihood of an animal being classified as a mammal. For example, the presence of the "milk" ($f_{milk}(1)$) strongly supports mammal classification as it's value is positive. However, the presence of "feathers" ($f_{feathers}(1)$) lowers the predicted probability of being a mammal. Also, the magnitude of these values reflects absolute feature importance for prediction of specific class: larger absolute values correspond to greater influence on the prediction. As shown in Figure 2, the presence of "milk" $f_{milk}(1)$ has the highest absolute contribution, making it the most important feature for identifying mammals. This aligns with biological knowledge of mammalian traits.

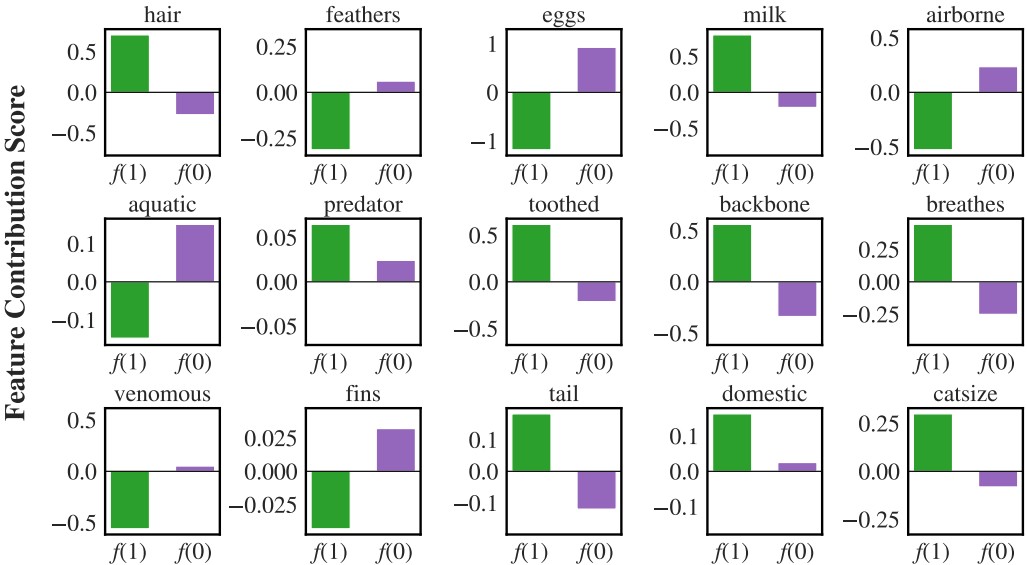

Figure 2: Feature contribution scores for predicting the "Mammal" class in the Zoo dataset. Each barplot shows the effect of a binary feature on the final prediction, where $f(1)$ represents the contribution when the feature is present and $f(0)$ when it is absent.

**Hyperedge-level interpretation.** HGNAN-edge can provide hyperedge-level interpretation for hyperedge prediction. In this setting, the combination of distance functions $\{\rho_s(\cdot)\}_{s=1}^{s_{max}}$ and the contribution of each feature, represented by $w_k f_k$ could fully explain the model's prediction. We use the iAF1260b dataset from the BiGG database for illustration, where each node feature corresponds to a certain function group. As mentioned in Section 3.3, we generate an embedding for each hyperedge by using difference pooling of node features on

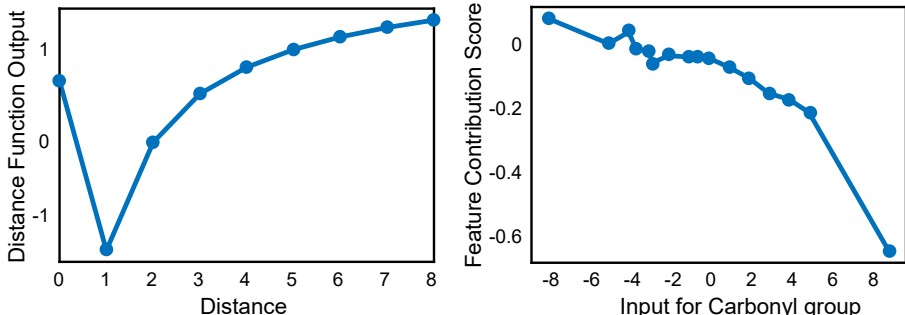

Figure 3: Visualization of the distance function (left) and Carbonyl group contribution score (right) for the iAF1260b dataset. Linear interpolation is used to connect the points.

the hyperedge. Therefore, the hyperedge embedding represents the difference in the number of functional groups before and after the reaction.

Figure 3 shows the distance function (left) and contribution of the Carbonyl group (C=O) to prediction (right). The distance function $\rho_1(\cdot)$ indicates structural information from the hypergraph. For this dataset, the optimal $s_{max}$ is 1. Therefore we just provide one distance function ($\rho_1(\cdot)$). The left subplot in Figure 3 shows that the model exhibits a V-shaped pattern. Features associated with the node on the same hyperedge (distance = 0) positively influence prediction, while features from immediate one-hop neighbors (distance = 1) have a strong negative effect. As the distance increases further, the negative influence diminishes and eventually becomes increasingly positive. This curve is a reasonable global kernel for representation learning and could be justified from biological perspective. At distance 0 (same reaction) stoichiometric coupling in steady state makes participants co-vary, justifying a large positive weight. Distance 1 often captures branch-point competition for a limited precursor; increasing flux down one branch reduces flow down the other, motivating a negative weight. At distance 2 this competitive effect is indirect and attenuated. Beyond two hops, local trade-offs diminish and pairs are more constrained by global objectives (e.g., biomass composition), so average associations become weakly positive, warranting positive weights.

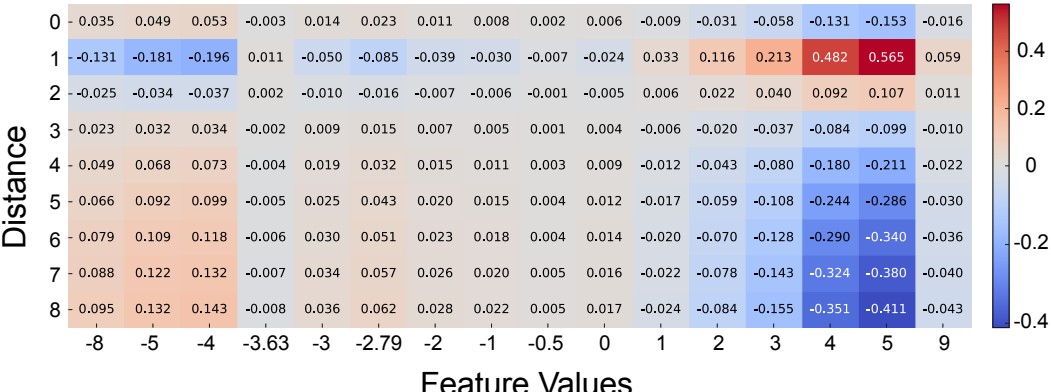

Figure 4: Heatmap for Carbonyl group: the horizontal axis stands for different feature value inputs $[\mathbf{y}_j]_{C=O}$ and the vertical axis stands for different distance inputs $d_1(e_i, e_j)$.

The term $w_k f_k$ quantifies how feature $k$ influences the model's output. The right subplot shows how the number of Carbonyl group (C=O) affect the prediction. The feature has a minor impact when the input of function group "C=O" is below 0, but its influence increases when $k$ is above 0. It means that generating more Carbonyl group after the reaction would have a huge impact on prediction. To compute the overall contribution to prediction, we have to combine the feature contribution ($w_k \cdot f_k$) with the structural weight from the distance

function ($\rho_1(\cdot)$). We visualize the combined contributions using heatmaps, as shown in Figure 4. Each cell in the heatmap represents the contribution of a particular feature-distance combination. For example, when a reaction results in the loss of 4 Carbonyl groups, the corresponding contribution is likely to be positive, as most values in the column of -4 are positive. These explanations allow domain experts to apply their knowledge to debug the model. For example, one can manually change the output of the distance function (e.g. $\rho_1(1)$) if it is not aligned with their domain knowledge and immediately get the prediction without retraining the whole model. It can also reveal some biologically meaningful patterns learned from the data, which may be valuable for downstream tasks.

## 5 Conclusion

HGNAN provides an inherently interpretable model where the decision-making mechanism can be visualized by 2D plots. In the meantime, HGNAN can achieve competitive or superior performance compared to state-of-the-art hypergraph learning methods across various node and hyperedge prediction tasks. This dual capability of delivering strong predictive performance while offering transparent decision-making makes HGNAN a valuable tool for scientific discovery in hypergraph-based applications, especially in fields such as biotechnology, bioinformatics, and social network analysis. One limitation, however, is that HGNAN constructs a separate neural network for each feature, which can lead to high computational costs on datasets with very large dimensionality. Fortunately, such extreme high-dimensional datasets are relatively uncommon in many real-world applications. Looking forward, future work will explore extensions to dynamic hypergraphs, further broadening its applicability and scalability.

## Reproducibility and Ethic Statement

An anonymous code is available at `https://anonymous.4open.science/r/HGNAN-6029`. We strictly adhere to the ICLR Code of Ethics ( `https://iclr.cc/public/CodeOfEthics`).

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
