# A    ADDITIONAL DETAILS FOR NODE CLASSIFICATION DATASETS

We provide detailed information about the node classification datasets in this appendix.

Each node in the Mushroom and Zoo datasets represents a mushroom or an animal, respectively mus (1981); Forsyth (1990). Hyperedges connect samples that share the same feature values, where the features describe the characteristics of each sample and are usually binary or categorical. The target variable for the Mushroom dataset is whether a mushroom is edible or not, while in the Zoo dataset, we predict the type of animal. The NTU2012 dataset Chen et al. (2003) is a 3D shape dataset consisting of 2012 objects across 67 distinct categories (e.g., cars, chairs, chessboards, clocks). Each node represents an object and is described by multi-view visual descriptors. Twitch-Gamers dataset requires binary classification of explicit-content presence per channel with Twitch accounts as nodes and accounts co-created within the same time window as hyperedges. Features include view counts, creation/update timestamps, language, lifetime activity duration, and an inactive flag. Pokec is a binary classification dataset of user gender. Nodes are users while each hyperedge is a user's complete friend set. Features span profile attributes such as age, hobbies/interests, education level, region, and registration time. Actor is a multi-class classification dataset of production role (actor/director/writer). Nodes are film-industry people while each hyperedge contains all collaborators on a single film. Features there are keyword-based attributes extracted from Wikipedia. Table 3 provides a more detailed summary about each dataset.

Table 3: Summary of node classification datasets. $|E|$ stands for the number of nodes on a hyperedge $E$ and $d_v$ stands for the number of hyperedges that contains a node $v$.

|             | NTU2012 | Mushroom | Zoo  | Pokec | Actor |
|-------------|---------|----------|------|-------|-------|
| $|V|$       | 2012    | 8124     | 101  | 3200  | 15761 |
| $|E|$       | 2012    | 298      | 43   | 2406  | 10164 |
| # feature   | 100     | 22       | 16   | 65    | 50    |
| # class     | 67      | 2        | 7    | 2     | 3     |
| max $|E|$   | 5       | 1808     | 93   | 7     | 28    |
| min $|E|$   | 5       | 2        | 2    | 2     | 1     |
| avg $|E|$   | 4.0     | 29.5     | 64.5 | 2.3   | 5.3   |
| max $d_v$   | 5       | 1808     | 61   | 7     | 205   |
| min $d_v$   | 1       | 1        | 1    | 1     | 1     |
| avg $d_v$   | 2.2     | 1.8      | 27.2 | 1.7   | 3.4   |

# B  ADDITIONAL DETAILS FOR HYPEREDGE PREDICTION DATASETS

The BiGG (Biochemical Genetic and Genomic) dataset King et al. (2016) is a repository of genome-scale metabolic models (GEMs) for various organisms, including bacteria, yeast, and human cells. It provides high-quality, standardized models in the *Systems Biology Markup Language* (SBML) format, facilitating metabolic network reconstruction and simulation. The BiGG dataset consists of multiple components. First, it includes genome-scale metabolic models that represent the biochemical processes of various organisms. It is stored as a stoichiometric matrix. These models define metabolic reactions, describing the biochemical transformations occurring within cells, and metabolites, which are the chemical compounds involved in these reactions. Additionally, the dataset provides gene-protein-reaction (GPR) associations, linking genes to their corresponding enzymes and metabolic functions.

**Feature generation.** BiGG models do not provide features for each metabolic. Common approaches for feature generation include using node2vec embeddings or extracting molecular fingerprints from SMILES. However, this method generate features that are not interpretable. In our paper, we generate feature for each metabolite using MACCS Keys. MACCS Keys are a type of molecular fingerprint that converts a molecule's structure into a 167-dimension binary vector. Each dimension indicates the presence or absence of a specific predefined substructure or functional group(i.e. F or carbon ring). Because the substructures are predefined by biologists, MACCS Keys offer a fast and interpretable way to vectorize molecules. However, since MACCS Keys are based on molecule fingerprints, which do not necessarily have one-to-one correspondence with each molecule, some molecules do not have MACCS-key-based features. For these molecules, we pad their features using 0. We also report this missing rate in Table 4.

**Access.** The BiGG dataset is publicly available at `http://bigg.ucsd.edu/`. MACCS Keys are available at `https://github.com/jAniceto/ml-knowledge-base/blob/main/02-data-preparation/feature-engineering/maccs.md`. Table 4 provides details about the four models from the BiGG dataset used in our paper.

Table 4: Summary of four datasets from the BiGG dataset.

|  | iAF1260b | iJR904 | iSB619 | iYO844 |
|---|---|---|---|---|
| $|V|$ | 2388 | 1075 | 743 | 1250 |
| $|E|$ | 1668 | 761 | 655 | 990 |
| Missing Rate | 0.764 | 0.912 | 0.829 | 0.840 |
| max $|E|$ | 67 | 56 | 61 | 63 |
| min $|E|$ | 1 | 1 | 1 | 1 |
| avg $|E|$ | 3.88 | 4.18 | 5.14 | 4.19 |
| max $d_v$ | 912 | 259 | 362 | 616 |
| min $d_v$ | 1 | 1 | 1 | 1 |
| avg $d_v$ | 5.55 | 4.77 | 5.83 | 5.29 |

To quantify homophily at different distances ddd on a hypergraph, we report the same-label ratio:

$$SLR(d) = \frac{1}{|V_d|} \frac{\sum_{j=1}^{N} \mathbb{1}(y_j = y_i)\mathbb{1}(\text{dist}(i,j) = d)}{\sum_{j=1}^{N} \mathbb{1}(\text{dist}(i,j) = d)}$$

where $V_d = \{i : \exists j \neq i, \text{dist}(i,j) = d\}\}$ is the set of nodes that have at least one neighbor exactly $d$ hops away. Since $SLR(d)$ can be misleading when the number of nodes at that distance is small, which is common when distance is large, we also provide coverage ratio across distances for reference.

For homophilic node classification datasets (e.g., NTU2012, Mushroom): $SLR(d)$ is high at small d and drops sharply with distance, indicating that near-hop neighborhoods are label-consistent while far-hop neighborhoods are not. In contrast, for heterophilic datasets (e.g., Pokec, Actor): there is no early decay. Actor shows a clear increase with distance; Pokec remains roughly flat around the mid-range hops and only declines later, consistent

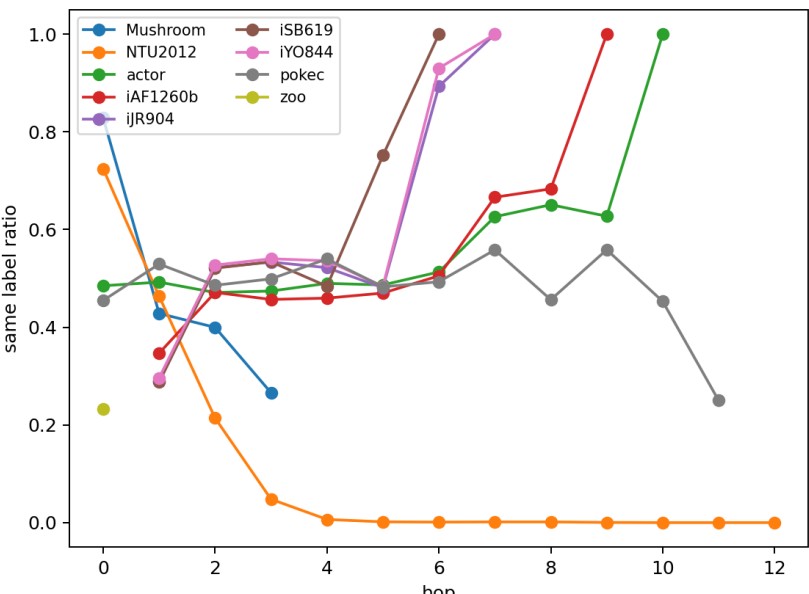

Figure 5: Same-label ratio by distance across all datasets

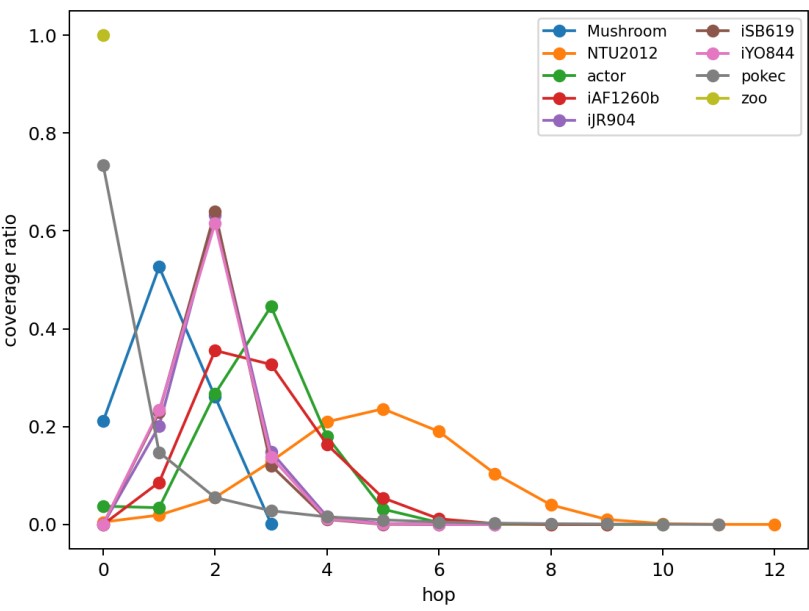

Figure 6: Coverage ratio by distance across all datasets

with weaker local homophily. For BiGG metabolic datasets (iAF1260b, iJR904, iYO844, iSB619), $SLR(d)$ increases with distance, often approaching 1 at larger hops, reflecting strong assortativity among far-hop nodes. Note that while far-hop $SLR(d)$ can be very high, the coverage at those hops is small, so the practical mass of labeled pairs still concentrates in the first few hops.

## C  DETAILS FOR MODEL TRAINING

In this appendix, we provide a detailed description of the experimental setup.

### C.1  NODE CLASSIFICATION TASKS

We perform grid search for hyperparameter tuning for each model to ensure optimal performance. We consider the following searching space for all methods: learning rate= $\{0.01, 0.001\}$, weight decay= $\{0, 0.0005\}$ and hidden channels= $\{64, 128, 256\}$. For methods using multi-head attentions, we tune the number of heads = $\{1, 4, 8\}$. For ED-HNN, we tune the number of layers = $\{0, 1\}$ and restart rate = $\{0, 0.5\}$. For our method, we also tune the number of layers = $\{3, 5\}$, dropout rate = $\{0, 0.5\}$ and $s_{\max} = \{1, 2, 3\}$. We select the hyperparameter configuration that achieves the best aggregated validation accuracy across all four datasets to guarantee robustness. The selected hyperparameters are reported in Table 5, and the corresponding evaluation result is in Table 1.

Table 5: Fixed best hyperparameters for node classification tasks

| Method | lr | wd | dropout | n_layer | hidden_channel | Classifier_hidden | heads | $s_{\max}$ | restart_alpha |
|---|---|---|---|---|---|---|---|---|---|
| HGNAN | 0.001 | 0 | 0.0 | 3 | 128 | - | - | 1 | - |
| AllDeepSets | 0.001 | 0 | - | - | 256 | 64 | - | - | - |
| AllSetTransformer | 0.001 | 0 | - | - | 256 | 64 | 8 | - | - |
| ED-HNN | 0.001 | 0 | - | 0,0,1 | 512 | 512 | - | - | 0.5 |
| HGNN | 0.01 | 0.0005 | - | - | 256 | 128 | - | - | - |
| HyperGCN | 0.01 | 0.0005 | - | - | 64 | 128 | - | - | - |
| UniGCNII | 0.001 | 0 | - | - | 64 | 256 | 8 | - | - |

Following Chien et al. (2021) and Wang et al. (2023a), we also select the best-performing hyperparameter configuration for each dataset individually. The optimal hyperparameters for each method–dataset combination and their corresponding performances on test set are reported in Table 6. HGNAN-node achieves accuracy comparable to that of baseline methods on most datasets. All experiments are run on NVIDIA V100 GPU.

Table 6: Best hyperparameters for each dataset for node classification tasks. The last column reports the test accuracy (mean ± standard deviation).

| Dataset | Method | lr | wd | dropout | n_layer | hidden_channel | Classifier_hidden | heads | restart_alpha | $s_{max}$ | Test Acc |
|---|---|---|---|---|---|---|---|---|---|---|---|
| NTU2012 | HGNAN | 0.001 | 0 | 0.5 | 3 | 256 | - | - | - | 3 | 0.896 ± 0.010 |
| NTU2012 | AllDeepSets | 0.001 | 0 | - | - | 256 | 64 | - | - | - | 0.878 ± 0.014 |
| NTU2012 | AllSetTransformer | 0.001 | 0 | - | - | 256 | 64 | 4 | - | - | 0.887 ± 0.010 |
| NTU2012 | HGNN | 0.01 | 0.0005 | - | - | 256 | 256 | - | - | - | 0.873 ± 0.014 |
| NTU2012 | HyperGCN | 0.01 | 0.0005 | - | - | 64 | 128 | - | - | - | 0.796 ± 0.033 |
| NTU2012 | UniGCNII | 0.001 | 0.0005 | - | - | 128 | 64 | 4 | - | - | 0.898 ± 0.015 |
| NTU2012 | ED-HNN | 0.001 | 0 | - | 0,0,1 | 512 | 256 | - | 0.5 | - | **0.899 ± 0.011** |
| Mushroom | HGNAN | 0.001 | 0 | 0 | 3 | 128 | - | - | - | 1 | **0.999 ± 0.001** |
| Mushroom | AllDeepSets | 0.01 | 0.0005 | - | - | 64 | 128 | - | - | - | **0.999 ± 0.001** |
| Mushroom | AllSetTransformer | 0.01 | 0 | - | - | 64 | 64 | 4 | - | - | **0.999 ± 0.001** |
| Mushroom | HGNN | 0.01 | 0.0005 | - | - | 256 | 256 | - | - | - | 0.998 ± 0.001 |
| Mushroom | HyperGCN | 0.001 | 0 | - | - | 64 | 64 | - | - | - | 0.482 ± 0.000 |
| Mushroom | UniGCNII | 0.001 | 0 | - | - | 64 | 128 | 1 | - | - | **0.999 ± 0.001** |
| Mushroom | ED-HNN | 0.001 | 0 | - | 0,1,1 | 512 | 128 | - | 0 | - | 0.998 ± 0.002 |
| zoo | HGNAN | 0.001 | 0 | 0 | 3 | 256 | - | - | - | 1 | 0.954 ± 0.033 |
| zoo | AllDeepSets | 0.001 | 0 | - | - | 256 | 64 | - | - | - | 0.942 ± 0.042 |
| zoo | AllSetTransformer | 0.001 | 0 | - | - | 64 | 64 | 1 | - | - | **0.973 ± 0.036** |
| zoo | HGNN | 0.01 | 0 | - | - | 64 | 128 | - | - | - | 0.954 ± 0.030 |
| zoo | HyperGCN | 0.001 | 0.0005 | - | - | 256 | 256 | - | - | - | 0.423 ± 0.000 |
| zoo | UniGCNII | 0.01 | 0 | - | - | 256 | 64 | 4 | - | - | 0.969 ± 0.016 |
| zoo | ED-HNN | 0.001 | 0 | - | 0,0,1 | 256 | 128 | - | 0 | - | 0.965 ± 0.028 |
| Actor | HGNAN | 0.001 | 0 | 0 | 3 | 64 | - | - | - | 1 | **0.863 ± 0.007** |
| Actor | AllDeepSets | 0.01 | 0 | - | - | 256 | 64 | - | - | - | 0.838 ± 0.003 |
| Actor | AllSetTransformer | 0.001 | 0.0005 | - | - | 256 | 64 | 8 | - | - | 0.836 ± 0.002 |
| Actor | HGNN | 0.01 | 0.0005 | - | - | 128 | 64 | - | - | - | 0.748 ± 0.003 |
| Actor | HyperGCN | 0.001 | 0 | - | - | 64 | 64 | - | - | - | 0.630 ± 0.000 |
| Actor | UniGCNII | 0.001 | 0 | - | - | 128 | 128 | 4 | - | - | 0.822 ± 0.003 |
| Actor | ED-HNN | 0.001 | 0 | - | 0,1,1 | 512 | 256 | - | 0 | - | 0.857 ± 0.005 |
| Pokec | HGNAN | 0.001 | 0 | 0 | 3 | 64 | - | - | - | 2 | **0.636 ± 0.010** |
| Pokec | AllDeepSets | 0.01 | 0.0005 | - | - | 128 | 256 | - | - | - | 0.578 ± 0.011 |
| Pokec | AllSetTransformer | 0.001 | 0.0005 | - | - | 256 | 64 | 8 | - | - | 0.564 ± 0.010 |
| Pokec | HGNN | 0.001 | 0 | - | - | 64 | 64 | - | - | - | 0.552 ± 0.020 |
| Pokec | HyperGCN | 0.01 | 0 | - | - | 256 | 256 | - | - | - | 0.534 ± 0.012 |
| Pokec | UniGCNII | 0.01 | 0.0005 | - | - | 64 | 64 | 8 | - | - | 0.573 ± 0.016 |
| Pokec | ED-HNN | 0.001 | 0 | - | 0,1,1 | 512 | 256 | - | 0.5 | - | 0.628 ± 0.017 |

## C.2 HYPEREDGE PREDICTION TASKS

Following Chen & Liu (2024), we consider the following configurations for all methods: learning rate = $\{0.001, 0.01\}$ and weight decay = $\{0, 0.0005\}$.

For HGNAN, we set dropout rate = $\{0.0, 0.5\}$, number of hidden channels = $\{32, 64, 128\}$ and $s_{max} = \{1, 2, 3\}$.

For CHESHIRE we set embedding dimension = $\{128, 256\}$, convolutional dimension = $\{64, 128\}$, Chebyshev polynomial order $k = \{3, 5\}$, and dropout probability $p = \{0.1, 0.2\}$.

For NHP, we tune the embedding dimension = $\{128, 256\}$ and convolutional dimension = $\{64, 128\}$.

For HyperSAGNN. we tune the embedding dimension = $\{128, 256\}$, convolutional dimension = $\{64, 128\}$, and number of attention heads = $\{1, 3\}$.

Meanwhile, we tune the hyperparameters separately for each dataset. The selected hyperparameters are reported in Table 7. We show the test accuracy in Table 2 and test AUC, AUPRC, and F1-score in Table 8. HGNAN-edge outperforms all baseline models on all evaluation metrics and datasets. All experiments are run on NVIDIA V100 GPU.

Table 7: Best hyperparameters for each dataset for hyperedge prediction tasks

| Dataset | Model | lr | wd | emb_dim | k | heads | dropout | layer | hidden_channel | $s_{max}$ | Test Acc |
|---------|-------|-----|-----|---------|---|-------|---------|-------|----------------|-----------|----------|
| iAF1260b | CHESHIRE | 0.001 | 0.0005 | 256 | 5 | - | 0.5 | - | 128 | - | $0.834 \pm 0.050$ |
| iAF1260b | NHP | 0.01 | 0.0005 | 128 | - | - | - | - | 64 | - | $0.732 \pm 0.076$ |
| iAF1260b | HyperSAGNN | 0.001 | 0.0005 | 256 | - | 4 | - | - | 128 | - | $0.730 \pm 0.075$ |
| iAF1260b | HGNAN | 0.001 | 0 | - | - | - | 0.5 | 3 | 32 | 1 | $\mathbf{0.935 \pm 0.069}$ |
| iJR904 | CHESHIRE | 0.001 | 0.0005 | 256 | 5 | - | 0.5 | - | 128 | - | $0.850 \pm 0.068$ |
| iJR904 | NHP | 0.001 | 0 | 1 | - | - | - | - | 128 | - | $0.690 \pm 0.090$ |
| iJR904 | HyperSAGNN | 0.001 | 0.0005 | 128 | - | 4 | - | - | 128 | - | $0.753 \pm 0.056$ |
| iJR904 | HGNAN | 0.001 | 0 | - | - | - | 0 | 5 | 128 | 2 | $\mathbf{0.959 \pm 0.018}$ |
| iSB619 | CHESHIRE | 0.001 | 0.0005 | 128 | 5 | - | 0.5 | - | 128 | - | $0.831 \pm 0.038$ |
| iSB619 | NHP | 0.001 | 0.0005 | 1 | - | - | - | - | 128 | - | $0.687 \pm 0.055$ |
| iSB619 | HyperSAGNN | 0.001 | 0 | 256 | - | 4 | - | - | 64 | - | $0.729 \pm 0.162$ |
| iSB619 | HGNAN | 0.001 | 0 | - | - | - | 0 | 5 | 64 | 1 | $\mathbf{0.970 \pm 0.018}$ |
| iYO844 | CHESHIRE | 0.001 | 0.0005 | 256 | 3 | - | 0.5 | - | 64 | - | $0.893 \pm 0.047$ |
| iYO844 | NHP | 0.001 | 0 | 1 | - | - | - | - | 128 | - | $0.747 \pm 0.043$ |
| iYO844 | HyperSAGNN | 0.001 | 0.0005 | 256 | - | 4 | - | - | 128 | - | $0.808 \pm 0.045$ |
| iYO844 | HGNAN | 0.001 | 0 | - | - | - | 0.5 | 5 | 128 | 2 | $\mathbf{0.937 \pm 0.058}$ |

Table 8: Additional testing metrics for hyperedge prediction tasks. Bold values highlight the best result for each dataset. HGNAN-edge outperforms all baseline models across all datasets.

| Model | iAF1260b | | | iJR904 | | | iSB619 | | | iYO844 | | |
|-------|----------|----------|--------|----------|----------|--------|----------|----------|--------|----------|----------|--------|
| | AUROC | AUPRC | F1 | AUROC | AUPRC | F1 | AUROC | AUPRC | F1 | AUROC | AUPRC | F1 |
| CHESHIRE | 0.7844 | 0.7339 | 0.7012 | 0.7802 | 0.7497 | 0.7514 | 0.7591 | 0.7309 | 0.7427 | 0.8312 | 0.7934 | 0.7918 |
| NHP | 0.7566 | 0.7323 | 0.7130 | 0.7312 | 0.6901 | 0.7003 | 0.7183 | 0.6872 | 0.6961 | 0.7791 | 0.7470 | 0.7402 |
| HyperSAGNN | 0.7831 | 0.7303 | 0.7000 | 0.7843 | 0.7529 | 0.7505 | 0.7548 | 0.7290 | 0.7337 | 0.8424 | 0.8084 | 0.8085 |
| HGNAN-edge | **0.9951** | **0.9961** | **0.9774** | **0.9990** | **0.9991** | **0.9889** | **0.9954** | **0.9957** | **0.9726** | **0.9717** | **0.9717** | **0.9574** |

## D  Experiment on Synthetic Dataset

To validate that our model could learn the complex feature shape and distance functions, we build a hypergraph-level classification dataset whose data-generating process is isomorphic to HGNAN-edge.

**Construction of the hypergraph.** The hypergraph is consistent of $E = 2C + L$ hyperedges. Hyperedge–hyperedge intersection graph $G$ is created with two sparse clusters of size $C$ each, stitched by a chain of length $L$ whose chain head connects to the left cluster, tail to the right. Each node of $G$ represents a hyperedge in the hypergraph. The node features $z_i \in \mathbb{R}^2$ are sampled i.i.d from $U(-1, 1)$. The hyperedge feature is the mean of its member node features: $x_e = \frac{1}{|V_e|} \sum_{i \in V_e} z_i$. The ground-truth mechanism is comprised of two fixed feature transforms and weights: $f_1(x_1) = -x_1, f_2(x_2) = \sin(\pi x_2)$ and $w_1 = 2, w_2 = 1$. The ground-truth distance function follows an exponential form: $\rho(d) = \exp(-\frac{1}{2}d)$. For each hyperedge, its logit is calculated using the same logic as (7). We standardize the logits using z-score and add Guassian noise $\mathcal{N}(0, 0.1)$ to it. It is then passed through sigmoid and threshold at 0.5 to produce hyperedge-level labels $y \in \{0, 1\}^E$.

**Measurements and Results**. To align the scales, we deploy an affined transformation to the results of our model. The coefficients of the transformation $a$ and $b$ are obtained by solving this simple linear regression: $\sum_i (af(\hat{x}_i) + b - f(x_i))^2$, where where $f$ is the ground truth function. We plot the comparison and report the MSE, $R^2$ and Pearson Correlation on the aligned curve. We can see that HGNAM-edge successfully recovered most of the underlying mechanism of this synthetic dataset.

Table 9: Alignment performance of HGNAM-edge compared with ground-truth. Lower MSE is better; higher $R^2$ and Pearson correlation are better.

| Method | MSE | $R^2$ | Pearson Corr. |
|--------|------|-------|---------------|
| Feature$_1$ | 0.0013 | 0.9899 | 0.9949 |
| Feature$_2$ | 0.0014 | 0.9943 | 0.9972 |
| Distance | 0.0005 | 0.9905 | 0.9952 |

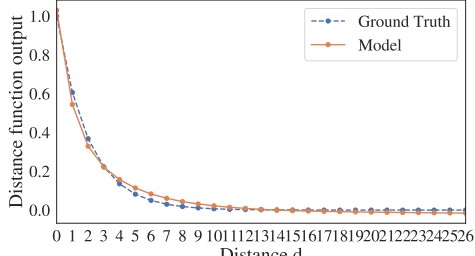

Figure 7: Distance function alignment

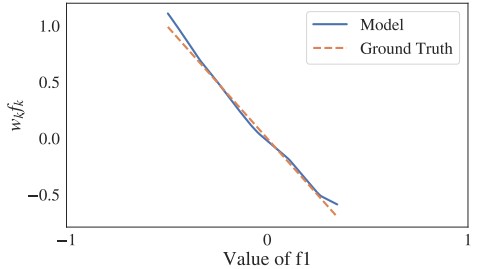

(a) Feature 1 shape function alignment

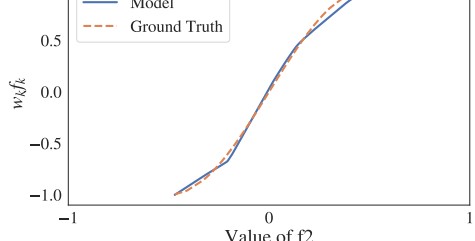

(b) Feature 2 shape function alignment

Figure 8: Feature shape function alignment

# E  ANALYSIS OF RELATIVE FEATURE IMPORTANCE

In Equation (4) and (7), we introduce a feature specific weight $w_k$ for each feature $k$. These weights are passed through an exponential function to ensure positive and finally normalized using softmax so that $\sum_{i=1}^{k} w_k = 1$. In this way, the model could learn the weights in the context of other features. In this way, we can interpret them as dataset-level relative feature importance.

Figure 9 reports global feature importance for predicting the seven classes in the UCI Zoo dataset, and the ordering is reasonable. The top variables—eggs, legs, backbone, breathes, and toothed—create coarse, high-information splits: eggs cleanly separates mammals from most other classes; legs distinguishes insects and some invertebrates from vertebrates ; backbone isolates invertebrates; breathes helps separate fish; and toothed differentiates mammals from birds. Mid-ranked features such as feathers, airborne, fins, aquatic, tail, and catsize are very informative but mainly within specific subsets (e.g., feathers/airborne nearly determine birds), so their global contribution is diluted. Milk or hair score lower because they are strongly correlated with the higher-ranked variables (e.g., eggs/toothed/backbone). Traits like predator and venomous are rare and cross multiple classes, so they naturally receive low importance. Overall, the ranking mirrors both the dataset's structure and known biases of global importance metrics, making the plot reasonable and interpretable.

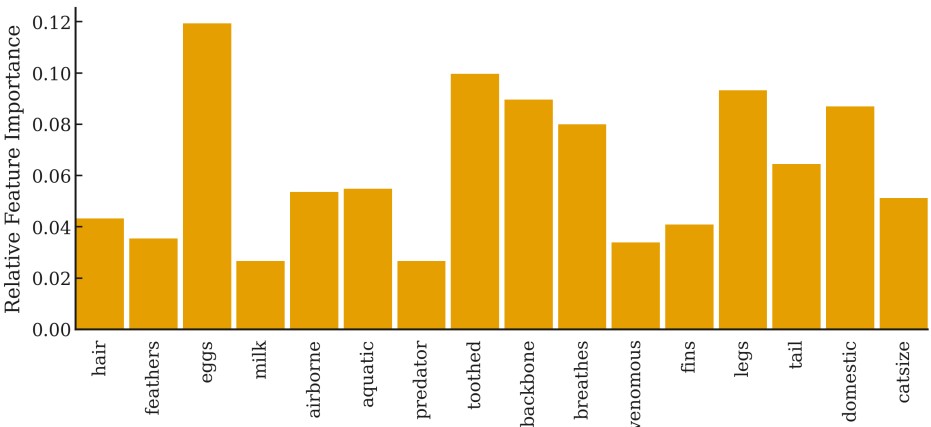

Figure 9: Relative feature importance $w_k$ for Zoo dataset

## F    Cross-dataset Validation for BiGG Datasets

We conduct cross-dataset validation to evaluate the model's ability to transfer knowledge across different GEM models. Specifically, we apply a model trained on one dataset (e.g., iAF1260b) to test on another. Each model is the best performer selected as described in Appendix C.2. After obtaining the best model trained on one dataset (e.g., iAF1260b), we fine-tune it for one additional epoch using the training set of a target dataset (e.g., iYO844). Since HGNAN is a NAM-based model, its feature-shape and distance functions maintain the same dimension across datasets, allowing direct transfer of the trained model to another dataset. Other baseline methods, however, may encounter input shape mismatches during this procedure. Therefore, for these methods, we first extract each dataset's incidence matrix, compute the union of all metabolites (i.e., nodes) across the four datasets, and then pad each incidence matrix with rows of value 0 for any metabolites that are absent. This ensures that all matrices have the same dimension, enabling fair cross-dataset validation.

According to Table 10, HGNAN outperforms baseline methods across almost all test settings. In one transfer setting, where the model trained on iAF1260b is applied to iJR904, HGNAN-edge achieves near-perfect classification performance with an AUCROC of 0.988. Even for the worst transfer, which is from iAF1260b to iYO844, the performance is above the average among all models. On average, HGNAN-edge achieves an AUROC of approximately 0.818 across different transfer settings, significantly outperforming baseline methods. CHESHIRE and NHP perform pooly on this task. While HyperSAGNN is competitive in some transfer settings, it has higher variance compared to HGNAN-edge. These results demonstrate that HGNAN-edge effectively captures the underlying metabolic network structure, enabling more robust transfer learning performance.

Table 10: Cross-dataset validation results for the BiGG datasets. AUROC is used as the performance metric. Bold values highlight the best result for each pair of transfers.

| Training Data | Testing Data | NHP | HyperSAGNN | CHESHIRE | HGNAN-edge |
|---|---|---|---|---|---|
| iAF1260b | iJR904 | 0.6213±0.4679 | 0.7769±0.2035 | 0.6785±0.4400 | **0.9884±0.0216** |
| iAF1260b | iYO844 | 0.6284±0.3968 | **0.7479±0.2502** | 0.5059±0.3897 | 0.6254±0.1364 |
| iAF1260b | iSB619 | 0.6460±0.2731 | 0.7445±0.3905 | 0.5455±0.5925 | **0.8297±0.1204** |
| iJR904 | iAF1260b | 0.6476±0.1190 | 0.7657±0.0308 | 0.6717±0.2569 | **0.8679±0.3468** |
| iJR904 | iYO844 | 0.6667±0.3766 | 0.7641±0.3306 | 0.6406±0.1578 | **0.8967±0.1469** |
| iJR904 | iSB619 | 0.6291±0.2954 | **0.7137±0.4300** | 0.5017±0.3161 | 0.6652±0.1125 |
| iYO844 | iAF1260b | 0.6551±0.1080 | 0.7627±0.4770 | 0.6081±0.5619 | **0.9624±0.0153** |
| iYO844 | iJR904 | 0.6143±0.5227 | **0.8191±0.1564** | 0.6637±0.5340 | 0.6534±0.0866 |
| iYO844 | iSB619 | 0.6420±0.2554 | **0.7950±0.5768** | 0.5888±0.2448 | 0.7216±0.2165 |
| iSB619 | iAF1260b | 0.6754±0.4719 | 0.7333±0.3460 | 0.5827±0.4503 | **0.8345±0.3116** |
| iSB619 | iJR904 | 0.5798±0.0419 | 0.8332±0.4215 | 0.5981±0.1856 | **0.8661±0.3416** |
| iSB619 | iYO844 | 0.6598±0.2153 | 0.8218±0.5518 | 0.6404±0.1902 | **0.9013±0.3363** |

## G  NODE-NEIGHBOR LEVEL EXPLANATION

In Section 3.2, we showed that HGNAN-node learns a GAT-style small attention network to assign weights $a_{ij}$ to each neighbor $j \in \mathcal{N}_i^s$ of each node $i$. It is also trained with $\ell_1$ norm to encourage neighbor-level sparsity and keep only important neighbors. To illustrate the node-level interpretability obtained by HGNAN-node, we provide a small example on the NTU2012 dataset.

We randomly selected two nodes in the hypergraph, investigating the neighbors that partici-pating in the decision making process for the target node. Figure 10 (for node 32) and 11 (for node 1036) give examples for visualization of the interpretation. In each plot, the red star denotes the target node, and the highlighted neighbors (with black outline) correspond to those receiving attention weights. From the visualization, we can clearly see which neighbor contributes to the target node's prediction while which neighbors do not.

For both nodes, HGNAN-node assigns non-negligible attention to only a very small subset of neighbors (3 out of 28 for node 32; 3 out of 64 for node 1036), demonstrating the intended sparsity behavior. Notably, these influential neighbors share the same class label as the target node, despite the fact that many other neighbors carry different labels. This indicates that the model is able to distinguish informative neighbors from numerous noises.

In addition, although the NTU2012 hyperedges contain many nodes with homogeneous labels, the selected influential neighbors often come from non-local or higher-order connections rather than trivial immediate ones. These examples suggest that HGNAN-node leverages higher-order hypergraph structure while maintaining clear and interpretable neighbor-level reasoning.

## H  LLM USE DECLARATION

In this paper, we used LLM to help us only with polishing our writing.

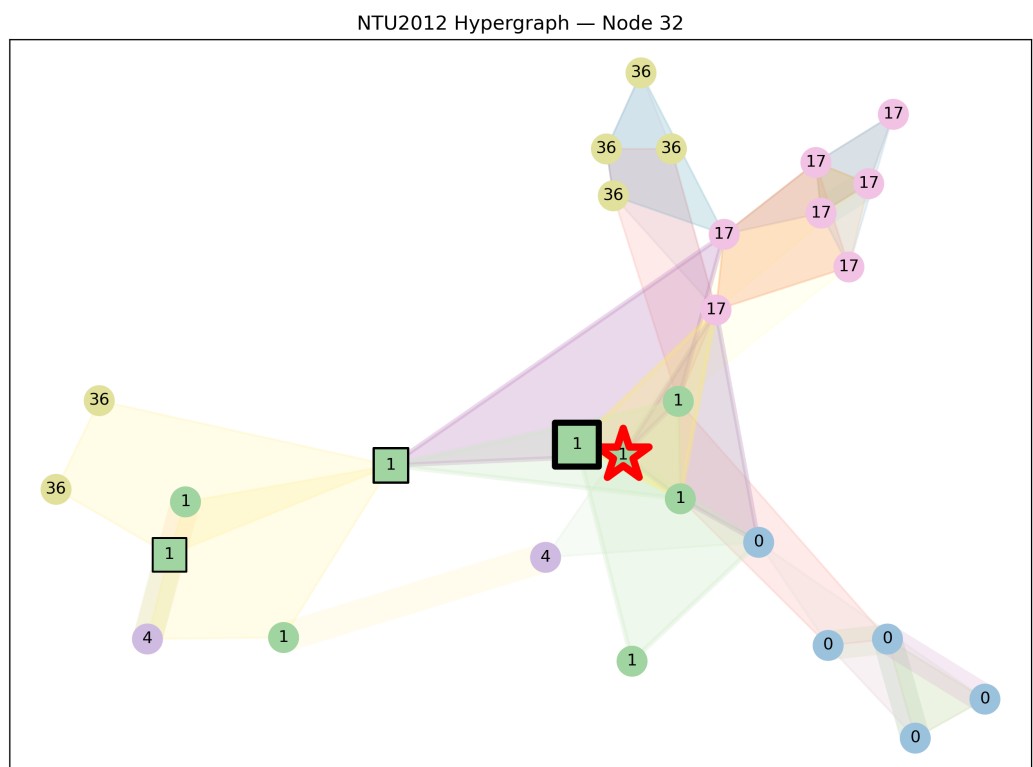

Figure 10: Node-level interpretation example for node 32 in the NTU2012 dataset. The red star highlights the target node we are interested in. Square nodes with black outline are important neighbor for explaining the prediction node of the target node. The thicker the outline, the more important the neighbor is. Number for each node denotes the ground-truth label for the node. Each class is labeled with unique colors. Different hyperedges are also colored differently. Neighbors within 2 hops are visualized.

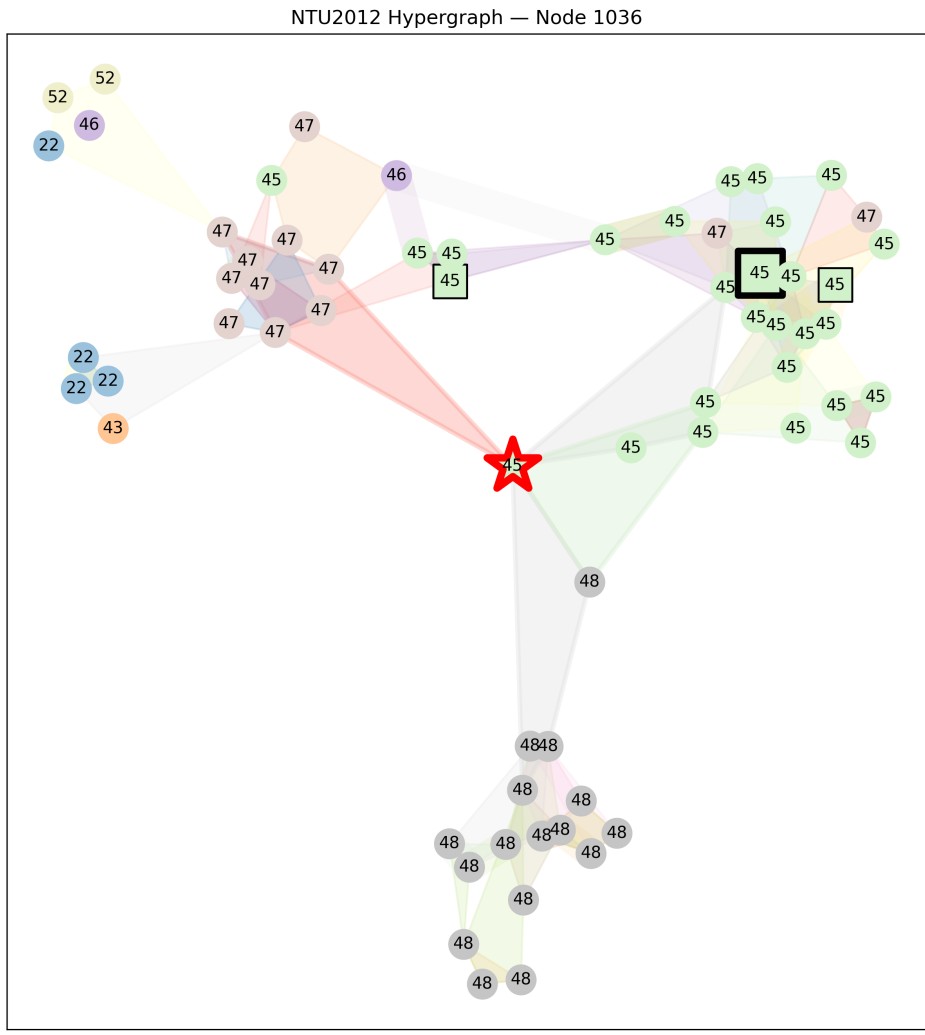

Figure 11: Node-level interpretation example for node 1036 in the NTU2012 dataset. The red star highlights the target node we are interested in. Square nodes with black outline are important neighbor for explaining the prediction node of the target node. The thicker the outline, the more important the neighbor is. Number for each node denotes the ground-truth label for the node. Each class is labeled with unique colors. Different hyperedges are also colored differently. Neighbors within 2 hops are visualized.