# OpenReview forum: "Interpretable Hypergraph Neural Additive Networks"
_ICLR.cc/2026/Conference — Submitted to ICLR 2026_

### Official Review · Reviewer_gAzN · 2025-10-27

**Soundness:** 3
**Presentation:** 2
**Contribution:** 2
**Rating:** 4
**Confidence:** 3

**Summary:**

This paper introduces HGNAN, a hypergraph neural additive network designed to achieve inherent interpretability in hypergraph learning. Unlike conventional black-box hypergraph neural networks that rely on post-hoc explanations, HGNAN extends generalized additive models to provide transparent and visualizable decision mechanisms while maintaining strong expressive capability. Experimental results across multiple node classification and hyperedge prediction tasks show that HGNAN achieves competitive or superior performance compared to state-of-the-art methods, and offers interpretable insights in scientific domains such as bioinformatics and social network analysis.

**Strengths:**

S1: Extending additive networks to hypergraph learning is an interesting and meaningful direction, and the proposed model demonstrates a certain degree of interpretability.

S2: The experimental results are comprehensive and show a good balance between predictive performance and interpretability.

**Weaknesses:**

W1: The overall writing quality needs improvement, as several parts of the method and experiments are not clearly described. For example, the methodology section only explains how node or hyperedge representations are obtained, but the complete training pipeline and model architecture are not well presented. In addition, some key parameters, such as $W$, are not properly introduced—their dimensions and correspondence with samples remain unclear.

W2: The paper primarily extends the GNAN framework to hypergraphs, with the main difference appearing to be the definition of a distance function on hypergraphs. However, the overall idea is largely similar to GNAN, suggesting that the methodological novelty may be limited.

W3: Although the authors mention that “such extreme high-dimensional datasets are relatively uncommon in many real-world applications,” in practice, many graph datasets feature high-dimensional node attributes, especially those derived from large language model embeddings. This could limit the applicability and scalability of the proposed method.

W4: The paper does not provide any theoretical analysis of time complexity or experimental evaluation of computational cost. It is recommended that the authors include such analyses to strengthen the work.

**Questions:**

Q1: I would like to know the specific values of $s_{max}$ used for each dataset during the experiments, as this information does not seem to be provided in the paper.

Q2: I am curious about how the results in Figure 2 were computed. In particular, is the weight vector $W$ introduced in Section 3.2 shared across the entire dataset, or is it computed separately for each sample?

---

> ### Author Response · Authors · 2025-11-21
>
> We thank the reviewer for their thoughtful comments. Below we address the reviewer’s concerns point-by-point.
>
> **W1. The overall writing quality needs improvement, as several parts of the method and experiments are not clearly described. For example, the methodology section only explains how node or hyperedge representations are obtained, but the complete training pipeline and model architecture are not well presented. In addition, some key parameters, such as W, are not properly introduced—their dimensions and correspondence with samples remain unclear.**
>
> We appreciate the reviewer’s comments on clarity, and we have revised the writing in Section 3 accordingly to make each notation clear. We also have included a small subsection describing our training pipeline (Section 3.4).
>
> **W2. The paper primarily extends the GNAN framework to hypergraphs, with the main difference appearing to be the definition of a distance function on hypergraphs. However, the overall idea is largely similar to GNAN, suggesting that the methodological novelty may be limited.**
>
> The connection between HGNAN and GNAN through the NAM-style additive formulation is intentional: our goal is to bring the benefits of Neural Additive Models to the hypergraph domain. However, we respectfully disagree that HGNAN is essentially GNAN with a different distance. There are several hypergraph-specific problems that GNAN does not address, and HGNAN introduces non-trivial architectural components to handle them.
>
> **1) What problem do we solve on hypergraphs?**
> Most existing HGNNs typically operate either on clique expansions of hypergraphs or on bipartite/star expansions, which tend to flatten higher-order topology into pairwise structures. This makes it difficult to explicitly reason about how hyperedges interact with each other at different scales. HGNAN is designed to make this hypergraph topology explicit and interpretable: we use $s$-adjacency for nodes and $s$-intersection graphs for hyperedges, together with NAM-style decomposition, so that both higher-order structure and feature effects can be attributed in a transparent way.
>
> **2) HGNAN-edge is more than GNAN with a new distance.**
> GNAN is defined on simple graphs and focuses on node-level explanations. In contrast, HGNAN-edge tackles a hyperedge-level prediction and explanation problem that GNAN does not cover. We first decompose the hypergraph into a family of hyperedge-hyperedge graphs given by the $s$-intersection construction (for $s = 1, \dots, s_{\max}$), where each graph encodes a different notion of “closeness” between hyperedges based on the size of their node intersections. We then work on the dual of the hypergraph: hyperedges become nodes in each $s$-intersection graph, and on each of these graphs we apply a GNAN-style additive architecture enhanced with a distance-aware kernel over graph distances. The resulting hyperedge representations from different $s$-levels are finally combined in an additive way.
>
> Thus, HGNAN-edge is not obtained by simply swapping the distance function inside GNAN. It (i) introduces a hypergraph-specific decomposition into multiple $s$-intersection graphs, (ii) operates on the dual hyperedge graph to convert a hyperedge-level task into a sequence of node-level tasks on these derived graphs, and (iii) aggregates across $s$ to capture multi-scale high-order hyperedge interactions. This design enables hyperedge-distance explanations: for a given hyperedge, we can attribute its prediction to other hyperedges at different distances and intersection levels. It is something GNAN does not address.
>
> **3) HGNAN-node differs fundamentally from GNAN.**
> HGNAN-node also goes beyond directly applying GNAN to a hypergraph Laplacian. Instead of using a distance kernel as in GNAN, HGNAN-node defines node $s$-adjacency from the hypergraph incidence structure and employs a neighbor-level aggregation with GAT-style attention weights $a_{ij}$ and an $\ell_1$ regularizer to encourage sparsity over neighbors. Combined with the per-feature additive subnetworks, this yields node-neighborhood explanations on hypergraphs: for each node, we can identify which neighbors, at which $s$-levels, and through which features are most influential. **We have added an example for node-level interpretability in Appendix G.** GNAN neither models high-order structures nor provides such sparse, neighbor-level attribution on hypergraphs.
>
> In summary, while HGNAN is indeed inspired by NAMs and GNAN, it is not a trivial extension. It **(i) targets hypergraph topology explicitly through $s$-adjacency and $s$-intersection graphs, (ii) introduces a new hyperedge-level additive architecture based on dual $s$-intersection graphs, and (iii) replaces GNAN’s distance kernel with a hypergraph-specific, attention-based neighborhood aggregation in HGNAN-node.** HGNAN is the first inherently interpretable hypergraph neural network that provides intrinsic explanations at both the node and hyperedge levels.

---

> ### Author Response · Authors · 2025-11-21
>
> **W3. In practice, many graph datasets feature high-dimensional node attributes, especially those derived from large language model embeddings. This could limit the applicability and scalability of the proposed method.**
>
>
> High-dimensional node attributes do not imply a fundamental scalability issue for our method. NAM-based models allocate a small subnetwork per feature, and the resulting linear dependence on the feature dimension is a standard characteristic shared across NAM-based architectures to tradeoff for interpretability, not specific to HGNAN. Moreover, HGNAN can be combined with pre-hoc feature engineering to further reduce dimensionality when needed.
>
> Moreover, our work is primarily motivated by hypergraph applications where features are semantically meaningful, such as on biological and chemical hypergraphs, where the feature dimension is typically in the low hundreds (e.g., MACCS keys has 167 digits, RNA sequence descriptors typically has less than 1000 digits). In these settings, HGNAN’s linear-in-$p$ complexity remains practically manageable, and the per-feature shape functions provide interpretable, domain-aligned insights.
> In addition, in our model, each feature has a learnable weight $w_k$, and we can impose a $\ell_1$ penalty on w, which analogous to the neighbor-level sparsity we already use, to encourage feature-level sparsity, so that only a small subset of features receives weights.
>
> **W4. The paper does not provide any theoretical analysis of time complexity or experimental evaluation of computational cost. It is recommended that the authors include such analyses to strengthen the work.**
>
> We first provide a simple parameter complexity analysis. For HGNAN, we build a small MLP for each feature. So the total parameter count is linear in the feature dimension $p$. For comparison, in a standard HGNN, the hypergraph convolution is implemented as $H^{(l+1)} = \sigma\left(D_v^{-1/2} H W_e D_e^{-1} H^\top D_v^{-1/2} X^{(l)} W^{(l)}\right), where W^{(l)} \in \mathbb{R}^{p \times h} is a learnable linear map applied to node features. Since this $p \times h$ matrix dominates the parameter count, the total parameters also scale linearly with $p$ when $h$ is fixed. The main difference between HGNAN and HGNN is that we use many small subnetworks instead of a single shared projection matrix.
>
> To further validate our claim, we modify the Zoo dataset such that we copy the original features until the feature dimension reaches $P$. We vary $P$ from 100 to 2000, train HGNAN for one epoch and record the parameter size, training time and memory cost. Table 1 gives an overview of these efficiency metrics. These empirical results further show that HGNAN’s efficiency depends only linearly on the feature dimension $p$.
>
> | p    | trainable_params (M) | epoch_time (s) | peak_cuda_mem (MB) |
> |------|----------------------|--------------|------------------|
> | 100  | 1.786                | 0.216        | 50.47            |
> | 200  | 3.553                | 0.305        | 84.45            |
> | 300  | 5.320                | 0.449        | 118.43           |
> | 400  | 7.088                | 0.585        | 152.40           |
> | 500  | 8.855                | 0.718        | 186.38           |
> | 600  | 10.622               | 1.014        | 220.35           |
> | 700  | 12.389               | 1.004        | 254.33           |
> | 800  | 14.156               | 1.147        | 288.31           |
> | 900  | 15.924               | 1.286        | 322.28           |
> | 1000 | 17.691               | 1.434        | 356.26           |
> | 1100 | 19.458               | 1.571        | 390.23           |
> | 1200 | 21.225               | 1.895        | 424.21           |
> | 1300 | 22.992               | 1.860        | 458.19           |
> | 1400 | 24.760               | 2.005        | 492.16           |
> | 1500 | 26.527               | 2.154        | 526.14           |
> | 1600 | 28.294               | 2.474        | 560.11           |
> | 1700 | 30.061               | 2.450        | 594.09           |
> | 1800 | 31.828               | 2.800        | 628.07           |
> | 1900 | 33.596               | 2.727        | 662.04           |
> | 2000 | 35.363               | 3.064        | 696.02           |

---

> > ### Author Response · Authors · 2025-11-21
> >
> > **Q1: I would like to know the specific values of s_max used for each dataset during the experiments, as this information does not seem to be provided in the paper.**
> >
> > We have included it in the Appendix. For your reference, we listed here:
> > | Dataset    | NTU2012 | Mushroom | Zoo | Actor | Pokec | iAF1260b | iJR904 | iSB619 | iYO844 |
> > |------------|---------|----------|-----|-------|-------|----------|--------|--------|--------|
> > | s_max      | 3       | 1        | 1   | 1     | 2     | 1        | 2      | 1      | 2      |
> >
> > **Q2: I am curious about how the results in Figure 2 were computed. In particular, is the weight vector W introduced in Section 3.2 shared across the entire dataset, or is it computed separately for each sample?**
> >
> > Sorry for the confusion. In the text (line 398), we state that the plotted quantity is $w_k \times f_k$, but in the caption of Figure 2 we described it as only f_k by mistake. We clarify that the y-axis values in Figure 2 are the feature contribution scores, computed as the product of the global importance weight w_k and the corresponding feature shape value, and the x-axis values represent whether the feature is present or absent.
> > Regarding the reviewer’s question about \mathbf{w}: the weight vector $\mathbf{W} \in \mathbb{R}^p$ is shared across the entire dataset. It represents global relative feature importance and is not computed separately for each sample. For more detailed information, please refer to Appendix E.

---

### Official Review · Reviewer_Smkj · 2025-10-28

**Soundness:** 3
**Presentation:** 3
**Contribution:** 3
**Rating:** 6
**Confidence:** 4

**Summary:**

This paper introduces **HGNAN (Hypergraph Neural Additive Network)** — an *inherently interpretable* hypergraph neural network framework that integrates the additive modeling principle of **Neural Additive Models (NAMs)** with the structural learning capability of **Hypergraph Neural Networks (HGNNs)**. The key innovation lies in introducing **feature-wise additive shape functions** and **distance-based weighting mechanisms** that together preserve both *interpretability* and *higher-order relational reasoning*. Extensive experiments across **five node classification datasets** and **four hyperedge prediction datasets** demonstrate that HGNAN achieves *comparable or superior predictive accuracy* compared to black-box baselines such as HGNN, AllSetTransformer, and CHESHIRE, while providing interpretable insights at both the **feature** and **structure** levels.

**Strengths:**

* The work extends **Neural Additive Models** and **Graph Neural Additive Networks** to **hypergraph-structured data**, filling a notable gap in interpretable hypergraph learning.
* The concept of combining **distance-aware structural weighting** with **feature-wise additive modeling** is novel, enabling a direct interpretability mechanism rather than relying on post-hoc explainers.

* Experimental comparisons are comprehensive, covering diverse datasets that span homophilic and heterophilic regimes, ensuring that claims are empirically well supported.
* The interpretability visualizations (e.g., distance-function curves and feature contribution heatmaps) demonstrate genuine transparency into model reasoning.

* The paper is well-organized and readable, with clear notations and step-by-step explanations of key equations.

**Weaknesses:**

* Since HGNAN assigns a separate neural subnetwork to each feature (as in NAMs), scalability may become an issue for datasets with very high-dimensional feature spaces. Although the paper acknowledges this limitation, empirical evidence on runtime or parameter scaling is limited.

* A baseline for hypergraph node classification is missed[1].

[1] From Hypergraph Energy Functions to Hypergraph Neural Networks

**Questions:**

see weakness.

---

> ### Author Response · Authors · 2025-11-21
>
> We thank the reviewer for their thoughtful comments. Below we address the reviewer’s concerns point-by-point.
>
> **W1. Since HGNAN assigns a separate neural subnetwork to each feature (as in NAMs), scalability may become an issue for datasets with very high-dimensional feature spaces. Although the paper acknowledges this limitation, empirical evidence on runtime or parameter scaling is limited.**
>
> We first provide a simple parameter complexity analysis. For HGNAN, we build a small MLP for each feature. So the total parameter count is linear in the feature dimension $p$. For comparison, in a standard HGNN, the hypergraph convolution is implemented as $H^{(l+1)} = \sigma\left(D_v^{-1/2} H W_e D_e^{-1} H^\top D_v^{-1/2} X^{(l)} W^{(l)}\right)$, where $W^{(l)} \in \mathbb{R}^{p \times h}$ is a learnable linear map applied to node features. Since this $p \times h$ matrix dominates the parameter count, the total parameters also scale linearly with $p$ when $h$ is fixed.
>
> To further validate our claim, we modify the Zoo dataset such that we copy the original features until the feature dimension reaches $P$. We vary $P$ from 100 to 2000, train HGNAN for one epoch and record the parameter size, training time and memory cost. Table 1 gives an overview of these efficiency metrics. These empirical results further show that HGNAN’s efficiency depends only linearly on the feature dimension $p$.
>
> | p    | trainable_params (M) | epoch_time (s) | peak_cuda_mem (MB) |
> |------|----------------------|--------------|------------------|
> | 100  | 1.786                | 0.216        | 50.47            |
> | 200  | 3.553                | 0.305        | 84.45            |
> | 300  | 5.320                | 0.449        | 118.43           |
> | 400  | 7.088                | 0.585        | 152.40           |
> | 500  | 8.855                | 0.718        | 186.38           |
> | 600  | 10.622               | 1.014        | 220.35           |
> | 700  | 12.389               | 1.004        | 254.33           |
> | 800  | 14.156               | 1.147        | 288.31           |
> | 900  | 15.924               | 1.286        | 322.28           |
> | 1000 | 17.691               | 1.434        | 356.26           |
> | 1100 | 19.458               | 1.571        | 390.23           |
> | 1200 | 21.225               | 1.895        | 424.21           |
> | 1300 | 22.992               | 1.860        | 458.19           |
> | 1400 | 24.760               | 2.005        | 492.16           |
> | 1500 | 26.527               | 2.154        | 526.14           |
> | 1600 | 28.294               | 2.474        | 560.11           |
> | 1700 | 30.061               | 2.450        | 594.09           |
> | 1800 | 31.828               | 2.800        | 628.07           |
> | 1900 | 33.596               | 2.727        | 662.04           |
> | 2000 | 35.363               | 3.064        | 696.02           |
>
> **W2. A baseline for hypergraph node classification is missed: From Hypergraph Energy Functions to Hypergraph Neural Networks.**
>
> We agree that this method is a good baseline and have included it in our experiments. We have implemented it on the same node-classification benchmark datasets, using the same data splits and training protocol as for our other baselines. We have updated Table 1 in the paper. The table shows that PhenomNN performs competitively across datasets, particularly on Zoo, Mushroom and NTU2012, but fails to perform as well on heterogeneous hypergraphs such as Actor and Pokec. After adding the new baseline model, the conclusion remains the same: HGNAN can achieve competitive performance on most datasets and performs the best on the heterogeneous ones.

---

> ### Comment · Reviewer_Smkj · 2025-11-21
>
> Thank you for your responses. My concerns have been addressed and I will keep my score.

---

> > ### Author Response · Authors · 2025-11-21
> >
> > Thank you for your follow-up comment and for confirming that our responses addressed your concerns. We appreciate your time and constructive feedback.

---

### Official Review · Reviewer_ZmT6 · 2025-10-31

**Soundness:** 3
**Presentation:** 3
**Contribution:** 3
**Rating:** 8
**Confidence:** 3

**Summary:**

The paper presents an hypergraph neural network with an interpretable
mechanism (HGNAN), based on the additive architecture.

The efficacy of the new HGNAN is tested on node prediction and
hyper-edge prediction with results in line with the state of the art and
with the advantage of the interpretability mechanism that is clearly
explained in the experiments section.

The paper is well-written and accurate

**Strengths:**

The paper presents an innovative algorithm on hypergraphs and provides examples on node classification an hyperedge prediction

**Weaknesses:**

As stated by the authors a weakness can be the computational cost since a neural network is needed for each feature.

**Questions:**

Do the hypergraph-based techniques require a larger amount of input data compared to equivalent graphs?

---

> ### Author Response · Authors · 2025-11-21
>
> We thank the reviewer for their thoughtful comments. Below we address the reviewer’s concerns point-by-point.
>
> **W1. As stated by the authors a weakness can be the computational cost since a neural network is needed for each feature.**
>
> As we note in the paper, this is a conscious design choice to obtain a NAM-style glass-box architecture. This trade-off is inherent to NAM-based models in general. NAMs[1] and their graph extensions GNANs[2] also allocate such a separate subnetwork per feature to guarantee interpretability.
>
> In practice, this issue is not significant, as the overhead is limited for two reasons. First, each per-feature subnetwork $f_k$ is implemented as a very small MLP (3 layers with a small hidden size), so the total parameter count still remains lightweight and grows only linearly with the feature dimension $p$. In vanilla HGNN, the hypergraph convolution is implemented as $H^{(l+1)} = \sigma\left(D_v^{-1/2} H W_e D_e^{-1} H^\top D_v^{-1/2} X^{(l)} W^{(l)}\right)$, where $W^{(l)} \in \mathbb{R}^{p \times h}$ is a learnable linear map applied to node features. Since this $p \times h$ matrix dominates the parameter count, the total parameters also scale linearly with $p$ when $h$ is fixed.
>
> Second, in our target high-stakes applications (e.g., biological and chemical hypergraphs), the interpretable feature space is of moderate size (e.g., MACCS keys has 167 digits, RNA sequence descriptors typically has less than 1000 digits), making the linear scaling practically manageable. For truly high-dimensional raw features, HGNAN can be combined with pre-hoc feature engineering to further reduce cost without changing the core architecture.
>
> We also have conducted an additional experiment to further show that the computational cost is linear in $p$. We modify the Zoo dataset such that we copy the original features until the feature dimension reaches $P$. We vary $P$ from 100 to 2000, train HGNAN for one epoch and record the training time and memory cost. Table 1 gives an overview of these efficiency metrics.
> | p    | epoch_time(s) | peak_cuda_mem(MB) |
> |------|--------------|------------------|
> | 100  | 0.216        | 50.47            |
> | 200  | 0.305        | 84.45            |
> | 300  | 0.449        | 118.43           |
> | 400  | 0.585        | 152.40           |
> | 500  | 0.718        | 186.38           |
> | 600  | 1.014        | 220.35           |
> | 700  | 1.004        | 254.33           |
> | 800  | 1.147        | 288.31           |
> | 900  | 1.286        | 322.28           |
> | 1000 | 1.434        | 356.26           |
> | 1100 | 1.571        | 390.23           |
> | 1200 | 1.895        | 424.21           |
> | 1300 | 1.860        | 458.19           |
> | 1400 | 2.005        | 492.16           |
> | 1500 | 2.154        | 526.14           |
> | 1600 | 2.474        | 560.11           |
> | 1700 | 2.450        | 594.09           |
> | 1800 | 2.800        | 628.07           |
> | 1900 | 2.727        | 662.04           |
> | 2000 | 3.064        | 696.02           |
>
> **Q1. Do the hypergraph-based techniques require a larger amount of input data compared to equivalent graphs?
> To address the concern: our hypergraph-based methods do not require more input data than graph-based ones.**
>
> In all our experiments, the node (or hyperedge) feature matrix is exactly the same across graph and hypergraph models. The only difference lies in how the structural relationships are encoded. Graph models take an adjacency matrix $A \in \mathbb{R}^{n \times n}$ as input, while hypergraph models take an incidence matrix $H \in \mathbb{R}^{n \times m}$, where $H_{v,e}=1$ indicates that node $v$ participates in hyperedge $e$. In HGNAN, the node $s$-adjacency and hyperedge $s$-intersection graphs are deterministically derived from this incidence matrix $H$ during message passing. They do not require any additional samples or side information beyond what is already used to build $A$ or $H$.
>
> [1] Agarwal, Rishabh, et al. "Neural additive models: Interpretable machine learning with neural nets." Advances in neural information processing systems 34 (2021): 4699-4711.
>
> [2] Bechler-Speicher, Maya et al. “The Intelligible and Effective Graph Neural Additive Networks.” ArXiv abs/2406.01317 (2024): n. pag.

---

### Official Review · Reviewer_paWm · 2025-11-03

**Soundness:** 1
**Presentation:** 1
**Contribution:** 1
**Rating:** 2
**Confidence:** 5

**Summary:**

The paper introduces HGNAN, an interpretable hypergraph model that marries NAM-style per-feature shape functions with neighborhood-weighted aggregation for node tasks and distance-weighted aggregation over s-intersection graphs for hyperedge tasks. HGNAN can produce feature-level importance by decomposing the neural network to be a linear combination of NNs; each operates on a different feature independently.

**Strengths:**

- The problem itself is interesting, and the paper clearly motivates the need for inherently interpretable HGNNs.
- Some results on hyperedge prediction seem good.

**Weaknesses:**

- Novelty is limited. The paper only decomposes the HGNN into a linear combination of NNs. This is not new and can only produce feature-level explanations. Meanwhile, node/subgraph level explanations are still unknown.
- The expressive power of the HGNAN due to the linear decomposition is not discussed. What if we have many features? The number of parameters would grow linearly? What if those features are redundant or not linearly correlated? What is the expressiveness of HGNAN compared to other hypergraph neural networks?
- Evaluations are not on standard datasets.
- The paper does not use the standard font of ICLR.

**Questions:**

- The related work section should rather focus on interpretable by design GNN and post-hoc explainability for HGNN to better position itself in the literature.
- it may be better to start with a clear definition of the targeted explanation. This will better motivate architectural choices.
- The datasets used seem not to be standard. Can you reproduce the results used in other papers, such as cora, citeseer, pubmed, yelp, DBLP-CA?
- Feature explanations come from the linear combination of features are straightforward. Can you derive node explanation using HGNAN? For example, finding a set of nodes or hyperedges that are most influential to the prediction?

---

> ### Author Response · Authors · 2025-11-21
>
> We thank the reviewer for their thoughtful comments. Below we address the reviewer’s concerns point-by-point.
>
> **W1.& Q4. Novelty is limited. The paper only decomposes the HGNN into a linear combination of NNs. This is not new and can only produce feature-level explanations. Meanwhile, node/subgraph level explanations are still unknown.**
>
> We respectfully clarify that HGNAN is not a simple application of standard NAM on HGNN, nor is its interpretability limited to feature-level explanations. The core novelty lies in how additive modeling is reformulated for higher-order hypergraph structure, and how this leads to inherently interpretable node-level and hyperedge-level explanations, which are not available in prior work.
>
> **(1) HGNAN is not just a linear-combination decomposition of HGNN.**
> The vanilla HGNN [1] operates on weighted clique expansion of hypergraphs, whereas HGNAN truly learns the more complex structural information of the hypergraph.
> HGNAN leverages s-adjacency and s-intersection graphs and allows $s_{max}$ to be a tunable hyperparameter, enabling it to learn higher-order hypergraph structural information. HGNAN replaces the black-box transformations of traditional HGNNs with per-feature additive subnetworks, making interpretability an intrinsic part of its novelty.
>
> **(2) HGNAN can provide not only feature-level explanations, but also node-level and hyperedge-level explanations. In HGNAN-node, the explanations are defined at the neighborhood level, while in HGNAN-edge, they are based on distance-shape functions.**
>
> **a. HGNAN-node provides neighborhood-level explanations for each node**: In Eq. (2), the $k^{th}$ component of node $i$’s embedding under s-adjacency $[\mathbf{h}^s_i]\_k$ is computed with $a_{ij}$, which is a learnable weight that functions as a soft mask over the neighbors of $v_i$. Combined with the feature shape functions $f_k$​, this yields a node-neighborhood explanation: we can say “node $v_i$’s prediction is mainly driven by neighbors $v_{j_1}, v_{j_2},$ … through features $k_1, k_2,$ …”. We have updated the visualization for node level explanation in a new section in the Appendix G.
>
> **b. HGNAN-edge provides structure explanations based on the distance function**: In Eq. (6), we have a distance-based weighting function that captures the cumulative influence of hyperedges at varying distances from $e_i$ on the s-intersection graph. We have provided a more detailed analysis in Section 4.3.
>
> **W2. The expressive power of the HGNAN due to the linear decomposition is not discussed. What if we have many features? The number of parameters would grow linearly? What if those features are redundant or not linearly correlated? What is the expressiveness of HGNAN compared to other hypergraph neural networks?**
>
> Below we clarify why HGNAN remains expressive, efficient, and competitive, despite its additive decomposition.
>
> a) Expressiveness under linear decomposition
>
> HGNAN combines per-feature subnetworks with distance neural networks or neighborhood mask in order to gain interpretability. This is in line with the original Neural Additive Models (NAMs) [3] and their extensions on graphs (GNAN) [4], whose expressiveness has been repeatedly validated in diverse tabular and structured settings, without sacrificing inherent interpretability.
>
> In HGNAN, each $f_k$ is a non-linear MLP, and these non-linear transformations are further combined through attention-based neighborhood or distance-aware hyperedge aggregation and multi-scale $s$-intersection aggregation. Therefore, HGNAN can well capture non-linearity in structured hypergraph data.
>
> b) Feature dimension and parameter size
>
> The increase in parameter size with feature dimension $p$ is not specific to HGNAN; mainstream HGNN models also scale roughly linearly with $p$ because their learnable feature transformation matrices grow with the input dimension.
>
>  For example, in HGNN, the hypergraph convolution is implemented as $H^{(l+1)} = \sigma\left(D_v^{-1/2} H W_e D_e^{-1} H^\top D_v^{-1/2} X^{(l)} W^{(l)}\right)$, where $W^{(l)} \in \mathbb{R}^{p \times h}$ is a learnable linear map applied to node features. Since this $p \times h$ matrix dominates the parameter count, the total parameters scale linearly with $p$ when $h$ is fixed. In HGNAN, each per-feature subnetwork $f_k$ is a very small MLP (3 layers with a small hidden size), so the overall parameter count remains lightweight.
> Moreover, in our target high-stake setting where interpretability is most valued (such as biological or chemical hypergraphs like metabolic networks), the feature space is typically of moderate size (e.g., MACCS keys has 167 digits, RNA sequence descriptors typically has less than 1000 digits), where the linear scaling is practically manageable. In scenarios with very high-dimensional raw features, HGNAN can be combined with standard pre-hoc feature engineering before applying our interpretable hypergraph model.

---

> > ### Author Response · Authors · 2025-11-21
> >
> > c) Redundant or linearly correlated features
> >
> > In our aggregation, we impose a learnable feature weight $w_k$ for each feature-independent neural network. Then we can impose a $\ell_1$ penalty on $w_k$, analogous to the neighbor-level sparsity we already use, which encourages feature-level sparsity. In this case, only a small subset of features receives large weights. Also, we can select features based on the in-built relative feature importance $W$ for interpretation. Furthermore, we can use statistical tools (e.g. Variance Inflation Factor) to detect redundant or linearly correlated features and use pre-hoc feature engineering methods to drop them.
> >
> > d) Not linearly correlated features
> >
> > HGNAN currently focuses on the purely additive case to keep the architecture simple and the explanations maximally transparent. However, the idea of introducing a small number of pairwise feature interaction terms (similar to GA2M [2] and GAMI-Net [3]) could be naturally extended to hypergraph features, which we view as an exciting direction for future work. But our empirical results already indicate that the additive architecture of HGNAN is sufficiently expressive on the considered benchmarks.
> >
> > e) Comparison with existing HGNNs
> >
> > Our experiments have shown that HGNAN can outperform hyperedge prediction baselines on metabolic reaction prediction tasks and achieve on-par performance with baselines on node classification benchmark datasets.
> >
> > **W3.& Q3. Evaluations are not on standard datasets.**
> >
> > We first clarify that our evaluations are indeed performed on standard datasets.
> >
> > For the node classification experiments, we follow the widely adopted hypergraph benchmark introduced by [4], including Cora, Citeseer, Zoo, Mushroom, and NTU2012. In our work, we use a standard subset of these datasets, which have also been employed in many subsequent hypergraph learning papers built upon these benchmark datasets [5,6,7]. For the hyperedge prediction task, the BiGG metabolic models (e.g., iAF692, iAF1260b, iJO1366) are also widely used benchmarks in recent work [8,9,10].
> >
> > Moreover, this is precisely the kind of high-stakes, domain-driven application where interpretability is most desired. Our goal here is to demonstrate both that HGNAN can perform well on hyperedge prediction benchmarks and that its interpretability is not only of theoretical interest but also useful in a realistic biochemical setting.
> >
> > We also evaluate HGNAN-node on newly introduced heterogeneous benchmark datasets [11], where vertices with the same label are more evenly dispersed rather than clustered as in traditional homogeneous datasets (Figure 6, Appendix B). These settings prevent trivial local overfitting and require models to truly capture high-order hypergraph structure, making them substantially more challenging.
> >
> > In addition, we have conducted extra experiments on the Cora and Citeseer datasets as you mentioned. We have updated Table 1 in the paper, and the results further confirm that HGNAN is competitive with strong baselines on the most commonly used citation hypergraph benchmarks.
> >
> > **W4. The paper does not use the standard font of ICLR.**
> >
> > We apologize for the formatting oversight. This issue does not affect the technical content of the paper, and we have now updated our paper to use the correct ICLR font.
> >
> > **Q1. The related work section should rather focus on interpretable by design GNN and post-hoc explainability for HGNN to better position itself in the literature.**
> >
> > Our intention in the current draft was exactly to distinguish between (i) post-hoc explainability for (hyper)graph neural networks and (ii) interpretable-by-design additive models.
> >
> > Concretely, the paragraph titled “GNN explainability” discusses post-hoc GNN and HGNN explainers, including GNNExplainer, PGExplainer, XGNN, and the two hypergraph-specific methods HyperEX and SHypX, which all assume a pre-trained black-box model and then fit an additional explainer, which may not faithfully reflect the model's true reasoning. Immediately afterwards, the paragraph titled “Additive models” introduces inherently interpretable GAMs, NAMs, and Graph Neural Additive Networks (GNANs), highlighting the line of interpretable-by-design additive architectures.
> >
> > We agree that this structure can be made even clearer and more explicit. We have modified the Literature Review section according to your suggestions.

---

> > > ### Author Response · Authors · 2025-11-21
> > >
> > > **Q2. it may be better to start with a clear definition of the targeted explanation. This will better motivate architectural choices.
> > > 	We fully agree that clearly stating the targeted explanation would better motivate our architectural choices.**
> > >
> > > Our goal with HGNAN is to provide inherently decomposable explanations at three levels:
> > >
> > > (1) Global feature-level explanations: via per-feature shape functions $f_k(\cdot)$, which show how each input feature influences the prediction across its value range (similar to NAM/GAM plots).
> > >
> > > (2) Node-neighborhood explanations: in HGNAN-node, the sparse neighbor weights $a_{ij}$ identify, for a given node $v_i$, which neighbors $v_j$ and which feature(s) of neighbors are most influential for its prediction.
> > >
> > > (3) Hyperedge-distance explanations: in HGNAN-edge, distance-aware aggregation over the $s$-intersection graph attributes a hyperedge’s prediction to other hyperedges at different distances. It can reveal which structural neighborhoods in the hypergraph drive a given hyperedge prediction.
> > >
> > > We have modified our paper accordingly. You can find it in the Introduction section.
> > >
> > > [1] Feng, Yifan, et al. "Hypergraph neural networks." Proceedings of the AAAI conference on artificial intelligence. Vol. 33. No. 01. 2019.
> > >
> > > [2] Lou, Yin et al. “Accurate intelligible models with pairwise interactions.” Proceedings of the 19th ACM SIGKDD international conference on Knowledge discovery and data mining (2013): n. pag.
> > >
> > > [3] Yang, Zebin, Aijun Zhang, and Agus Sudjianto. "GAMI-Net: An explainable neural network based on generalized additive models with structured interactions." Pattern Recognition 120 (2021): 108192.
> > >
> > > [4] Chien, Eli, et al. "You are allset: A multiset function framework for hypergraph neural networks." arXiv preprint arXiv:2106.13264 (2021).
> > >
> > > [5] Kim, Jinwoo, et al. "Equivariant hypergraph neural networks." European Conference on Computer Vision. Cham: Springer Nature Switzerland, 2022.
> > >
> > > [6] Yu, Zhongming, et al. "Hypergef: A framework enabling efficient fusion for hypergraph neural network on gpus." Proceedings of Machine Learning and Systems 5 (2023): 387-399.
> > >
> > > [7] Telyatnikov, Lev, et al. "Hypergraph neural networks through the lens of message passing: A common perspective to homophily and architecture design." arXiv preprint arXiv:2310.07684 (2023).
> > >
> > > [8] Chen, Can et al. “Teasing out missing reactions in genome-scale metabolic networks through hypergraph learning.” Nature communications vol. 14,1 2375. 25 Apr. 2023, doi:10.1038/s41467-023-38110-7
> > >
> > > [9] Yang, Yang et al. “Lhp: Logical Hypergraph Link Prediction.” SSRN Electronic Journal (2022): n. pag.
> > >
> > > [10] Liu, Xiaoyi et al. “A generalizable framework for unlocking missing reactions in genome-scale metabolic networks using deep learning.” ArXiv abs/2409.13259 (2024): n. pag.
> > >
> > > [11] Lin, Junhong, et al. "When heterophily meets heterogeneity: New graph benchmarks and effective methods." arXiv preprint arXiv:2407.10916 (2024).

---

> ### Comment · Reviewer_paWm · 2025-11-21
>
> Thanks, authors, for the detailed response and additional experiments. However, I'm not convinced by your answer on W2.
>
> W2a) Expressiveness.
>
> If I understand correctly, this is no more than performing separate projections on each feature and then applying a linear regression on top of that to aggregate feature information. This architecture is quite restrictive in my opinion. Because, in a lot of scenarios, a feature itself does not contain much information and can also not be linearly correlated. Message passing among nodes with a single feature alone is also concerning, since there might not be much information to share among nodes with just a scalar.
>
> W2b) Feature dimension and parameter size
>
> I dont agree that the parameters of HGNN will increase linearly with the number of features. Your argument is correct for the first layer. In later layers, the number of parameters only depends on h, not p anymore.
>
> W2c) But you still need to create m networks for m redundant features?
>
> W2d) It seems that adding pairwise interaction networks will make the model even more cumbersome.
>
> For W1. Could you elaborate on what the core novelty is? What is challenging about applying NAM on HGNN that your solution addresses? What major differences are compared to vanilla NAM? Rather than each f is an HGNN applied to a single feature? What are the major observations that changed the current observations in the literature?

---

> ### Author Response · Authors · 2025-11-22
>
> We thank the reviewer for their comments. Below we address the reviewer’s additional concerns point-by-point.
>
> **W2a) Expressiveness. If I understand correctly, this is no more than performing separate projections on each feature and then applying a linear regression on top of that to aggregate feature information. This architecture is quite restrictive in my opinion. Because, in a lot of scenarios, a feature itself does not contain much information and can also not be linearly correlated. Message passing among nodes with a single feature alone is also concerning, since there might not be much information to share among nodes with just a scalar.**
>
> Each per-feature subnetwork $f_k$​ is a non-linear MLP, not a linear projection. In our implementation, $f_k:\mathbb{R} \to \mathbb{R}^{C}$ (where $C$ is the number of classes, or 1 for binary tasks) is a small MLP. Thus, for each feature $k$ and each node/hyperedge, HGNAN produces a vector-valued, non-linear transformation rather than a single scalar coefficient. These transformed features are then combined with attention-based neighborhood aggregation (HGNAN-node) or distance-aware $s$-intersection aggregation (HGNAN-edge). The model is additive across features by design, but each term is a rich, structure-aware component rather than “linear projection + linear regression”.
>
> Moreover, HGNAN explicitly exploits hypergraph structure to compensate for weak individual features. Even if each single feature $k$ for each node $i$, $x_{ik}$ is not informative on its own, its transformed value $f_k(x_{ik})$ is aggregated over the $s$-adjacency / $s$-intersection neighborhood via learnable attention weights $a_{ij}$ or distance kernels $\rho_s(d)$, so the contribution of feature $k$ to node/hyperedge $i$ depends on how this feature behaves in its local hypergraph context. This follows the same additive design used in NAMs and GNAN. Our experiments show that this expressiveness is sufficient in practice: HGNAN-node remains competitive with strong black-box HGNN baselines on node-level tasks, and HGNAN-edge significantly outperforms them on hyperedge prediction in the BiGG metabolic hypergraphs.
>
> **W2b) Feature dimension and parameter size. I don’t agree that the parameters of HGNN will increase linearly with the number of features. Your argument is correct for the first layer. In later layers, the number of parameters only depends on h, not p anymore.**
>
> We agree with the reviewer that, in a standard HGNN, only the first layer has parameters that explicitly depend on the input feature dimension $p$. But our intention was not to suggest otherwise, but rather to emphasize that the overall parameter count of HGNNs still scales linearly in $p$ due to this input layer when $h$ and the number of layers are fixed.
>
> For HGNAN, the situation is similar. Each per-feature subnetwork $f_k$ has a small and fixed architecture, so the total number of parameters in the feature-related part is approximately $p \cdot c_f$ for some constant $c_f$ determined by this architecture, plus additional structural parameters (attention and distance networks) that are independent of $p$. Thus, both HGNN and HGNAN have total parameter counts that grow linearly with the input feature dimension $p$. The difference lies in how this linear dependency is structured in the model (one large input matrix versus many small subnetworks), not in the order.
>
> **W2c)&d) But you still need to create m networks for m redundant features? It seems that adding pairwise interaction networks will make the model even more cumbersome.**
>
> Architecturally, HGNAN follows the standard NAM design in assigning one subnetwork $f_k$ to each input feature $x_{\cdot k}$. But when there is strong prior evidence that many features are redundant or highly correlated and we aim to keep the model small, HGNAN can be naturally combined with standard pre-hoc feature engineering, as is a common practice for GAMs/NAMs. If future applications reveal that stronger feature pairwise interactions are essential on certain hypergraph tasks (e.g. synergy between molecules), we can add a small set of feature pairwise interaction terms by deleting the redundant features and introducing a limited number of structured interaction terms with sparsity constraints.. It has been shown by GA2M and GAMI-Net to be able to further close the gap to black-box models without exploding model size. We view this as an interesting direction for follow-up work, but intentionally keep it out of scope here in order to keep the model simple and its explanations maximally transparent.

---

> > ### Author Response · Authors · 2025-11-22
> >
> > **W1. Could you elaborate on what the core novelty is? What is challenging about applying NAM on HGNN that your solution addresses? What major differences are compared to vanilla NAM? Rather than each f is an HGNN applied to a single feature? What are the major observations that changed the current observations in the literature?**
> >
> > HGNAN’s core novelty lies in these three aspects:
> >
> > **1. First interpretable-by-design hypergraph learning model.**
> >
> >    HGNAN is, to the best of our knowledge, the first hypergraph neural network that is glass-box by construction and provides explanations at the feature, node-neighborhood, and hyperedge-distance levels.
> >
> > **2. Non-trivial way of applying NAM to hypergraphs.**
> >
> > The challenge of applying NAM on HGNN lies in how to capture higher-order structure from hypergraphs while preserving its linear decomposable dependence on features. HGNAN decouples feature-wise non-linear transformation from hypergraph structure by linearly combining transformed features through a shared structural module based on $s$-adjacency (nodes) and $s$-intersection graphs (hyperedges). This design keeps complexity manageable while modeling higher-order hypergraph connectivity coherently, and preserving a clean additive decomposition across features, which goes beyond vanilla NAM. We also want to clarify that each $f_k$ is not an HGNN, but a lightweight small MLP.
> >
> > **3. Empirical observations on the interpretability-performance trade-off in hypergraphs.**
> >
> >   Our experiments suggest that HGNAN is an interpretable-by-design model while achieving on-par performance with black-box HGNNs, which changes the current observations that interpretable-by-design models usually trade performance for interpretability and fill in the gap in current literature that there is no existing interpretable-by-design hypergraph learning models.

---

> > > ### Comment · Reviewer_paWm · 2025-11-25
> > > **Thank you**
> > >
> > > Thank you again for the detailed responses. However, I remain unconvinced about the expressiveness of the proposed architecture, particularly in the high-dimensional feature setting.
> > >
> > > I understand that this may be a limitation of the broader line of work rather than this paper alone. Nevertheless, given the relatively limited novelty of the extension to HGNN, I am still inclined to recommend rejection. My impression is that Reviewer gAzN has raised similar concerns.
> > >
> > > That said, I have increased my score to a 4 to reflect the authors’ efforts in adding further experiments and improving the writing.

---

> > > > ### Author Response · Authors · 2025-11-25
> > > >
> > > > We thank the reviewers for the additional feedback and for increasing the score.
> > > >
> > > > We understand and respect the concern about expressiveness in very high-dimensional settings. Our goal in this work is to make a clear, interpretable-by-design trade-off. We deliberately enforce an additive structure as NAM and GNAN did to obtain faithful and decomposable explanations while retaining non-linear per-feature networks and hypergraph-structure-aware aggregation. Importantly, the experiments suggest that the expressiveness of HGNAN is sufficient on the hypergraph benchmark datasets. HGNAN-node remains competitive with strong black-box HGNN baselines on node tasks, while HGNAN-edge substantially outperforms them on BiGG hyperedge prediction and provides intrinsic interpretations. We will further clarify in the paper that extremely high-dimensional (e.g., LLM-generated embedding) settings are not the primary target use case, and discuss practical mitigation (e.g. feature selection or sparsity) and extensions such as introducing a small set of feature interaction terms as future work.

---

### Meta-Review · Area_Chair_APTU · 2026-01-06

**Summary:**

The authors introduce an inherently interpretable Hypergraph Neural Additive Network (HGNAN) for learning from higher-order relational data. Unlike existing hypergraph neural networks that rely on post-hoc explanations, HGNAN extends generalised additive models to provide transparent global and local interpretations at both node and hyperedge levels while retaining expressive power. Experiments on node classification and hyperedge prediction show competitive performance with state-of-the-art methods and improvements in recovering missing reactions in metabolic networks.

The reviews on the paper are mixed. Although the reviewers recognise the relevance of the topic and consider the authors' approach to be interesting and potentially useful, they nevertheless express a number of criticisms regarding novelty, expressiveness, evaluation, scalability and applicability, and lack of complexity analysis, among others. The authors could clarify some of the issues raised in the reviews, and the most critical reviewer was also willing to adapt their score, albeit still remaining on the negative side. Overall, the reviews are mostly borderline -- expect one very positive review, which, however is also very shallow and almost empty, and can hence hardly count. In conclusion, the paper does not meet the standards and high requirements of the conference in its current form and therefore cannot be recommended for acceptance.

**Reviewer Concerns:**

I would say that several concerns have been addressed, but some others remain including  limited novelty and the issue with expressivity.

**Reviewer Scores:**

The most negative reviewer raised the score from 2 to 4, another stays with 6. For the other 4, it's difficult to say whether or not they would have raised to 6. The reviewer with rating 8 should not count too much, I would say.

---

### Decision · Program_Chairs · 2026-01-26

Reject